# REINFORCEMENT LEARNING VIA VALUE GRADIENT FLOW

**Haoran Xu**[*1]    **Kaiwen Hu**[*2]    **Somayeh Sojoudi**[2]    **Amy Zhang**[1]
[1]University of Texas at Austin    [2]University of California, Berkeley

## ABSTRACT

We study behavior-regularized reinforcement learning (RL), where regularization toward a reference distribution (the dataset in offline RL or the base model in LLM RL finetuning) is essential to prevent value over-optimization caused by erroneous out-of-distribution extrapolation. Existing methods either rely on reparameterized policy gradient, which are difficult to scale to large generative models, or on reject sampling, which can be overly conservative when attempting to move beyond the behavior support. In this paper, we propose Value Gradient Flow (VGF), a scalable new paradigm for behavior-regularized RL. VGF casts behavior-regularized RL as an optimal transport problem that maps the reference distribution to the value-induced optimal policy distribution. We solve this transport problem via discrete gradient flow, where value gradients guide particles initialized from the reference distribution. Our analysis shows that VGF imposes regularization implicitly by controlling the transport budget. VGF eliminates explicit policy parameterization while remaining expressive and flexible, this enables adaptive test-time scaling by adjusting the transport budget. Extensive experiments demonstrate that VGF significantly outperforms prior methods, achieving state-of-the-art results on offline RL benchmarks (D4RL, OGBench) and LLM RL tasks.

**Code and runs can be found at https://ryanxhr.github.io/vgf**

## 1 INTRODUCTION

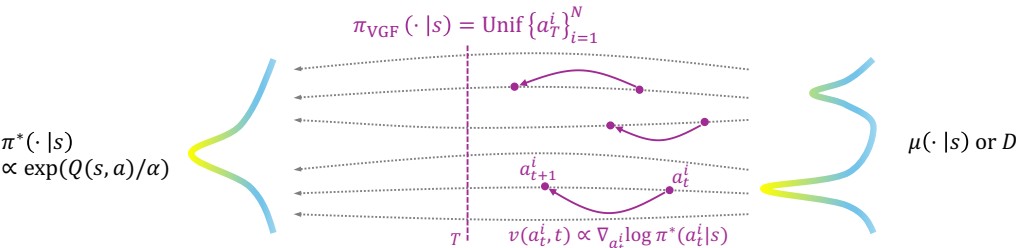

Figure 1: **VGF: Value Gradient Flow**. VGF reframes behavior-regularized RL as an optimal transport from the behavior distribution towards the Boltzmann value distribution, with the transport budget as implicit regularization. This scales to large generative models and enables adaptive test-time scaling.

Reinforcement learning (RL) has provided a powerful framework for solving sequential decision-making problems in complex environments. These methods have been successfully applied in diverse domains, ranging from robotics (Levine et al., 2016) to game playing (Mnih et al., 2013; Silver et al., 2017), and have recently become instrumental in fine-tuning large language models (LLMs) to align with human preferences and instructions (Ouyang et al., 2022) and enhancing the reasoning capabilities of LLMs (Shao et al., 2024; DeepSeek-AI et al., 2025). While these successes highlight the broad potential of RL, they also expose a key challenge: policies must often be regularized toward a reference distribution to remain stable and reliable. This challenge arises in both offline RL, where erroneous extrapolation beyond fixed datasets can cause severe value overestimation, and RLHF, where deviating too far from the supervised policy risks reward hacking. In these settings, naïve value maximization alone is insufficient. As a result, recent research in both offline RL (Kumar et al.,

---

[*]Equal contribution.

2020; Fujimoto et al., 2019; Wu et al., 2019; Xu et al., 2023) and RLHF (Ouyang et al., 2022; Wang et al., 2024) has increasingly converged on the paradigm of *behavior-regularized RL*, which balances value maximization with adherence to reliable reference distributions.

The most common approach to behavior-regularized RL is to add explicit divergence or distance penalties (e.g., L2 distance or KL divergence) to the RL learning objective (Ouyang et al., 2022; Touvron et al., 2023; Gao et al., 2023). While this constrains policies to remain close to the reference distribution, it also introduces several limitations. First, explicit penalty methods typically use a single coefficient to regularize both value learning and policy improvement, even though these two components can prefer different regularization strengths. Tying them together makes the penalty coefficient difficult to tune and can lead to either overly conservative updates or insufficient control of out of distribution drift (Wu et al., 2025; Korbak et al., 2022). Second, scaling these methods to expressive generative policies such as diffusion (Song et al., 2020; Ho et al., 2020) and flow models (Lipman et al., 2022) is challenging. Computing policy gradient requires differentiating through multi step sampling procedures, which is unstable and computationally expensive, while distillation the multi step policy into a one step model can sacrifice expressivity (Park et al., 2025b).

**Behavior-regularized RL without regularization.**[1]  We introduce **Value Gradient Flow (VGF)**, which casts behavior-regularized RL as the optimal transport from an (estimated) reference distribution to the optimal policy distribution induced by the value function. VGF adds no explicit distance or divergence penalty and does not rely on a parameterized policy. Instead, it starts from samples of the reference distribution and gradually nudges them toward higher value regions using a small, fixed number of guidance steps. The distribution after these steps serves as an implicit policy. The transport budget itself (i.e., how far and how often we move) acts as an implicit behavior regularization, limiting deviation from the reference distribution during training while preserving flexibility to enable adaptively scaling at inference. Notably, VGF supports different levels of regularization during training and inference. When the inference transport budget is zero, VGF reduces to reject sampling methods, Conversely, using a larger transport budget at inference than during training brings test-time-scaling improvements. In RLHF, VGF yields inference-time control that steers a supervised finetuned policy using first-order gradient, sidestepping RL-style optimization and reducing both compute and engineering complexity. Across extensive experiments, VGF consistently outperforms strong behavior-regularized baselines, attaining state-of-the-art results on standard offline RL suites (D4RL, OGBench) and delivering substantial gains on RLHF tasks.

## 2 PRELIMINARIES

**MDP and value functions.**  We consider the RL problem presented as a Markov Decision Process (MDP) (Sutton et al., 1998), which is specified by a tuple $\mathcal{M} = \langle \mathcal{S}, \mathcal{A}, \mathcal{P}, d_0, r, \gamma \rangle$. Here $\mathcal{S}$ and $\mathcal{A}$ are state and action space, $\mathcal{P}(s'|s,a)$ and $d_0$ denote transition dynamics and initial state distribution, $r(s,a)$ and $\gamma$ represent reward function and discount factor, respectively. The goal of RL is to find a policy $\pi(a|s)$ which maximizes expected return $J(\pi) = \mathbb{E}_\pi[\sum_{t=0}^\infty \gamma^t \cdot r(s_t, a_t)]$. In the offline setting, interaction with the environment is prohibited and one needs to learn an optimal $\pi$ from a static replay buffer $\mathcal{D} = \{s_i, a_i, r_i, s_i'\}_{i=1}^N$ collected from unknown policies. The dataset can be heterogeneous and suboptimal, we denote the empirical behavior policy of $\mathcal{D}$ as $\pi_{\mathcal{D}}$, which represents the conditional distribution $p(a|s)$ observed in the dataset.

RL methods based on approximate dynamic programming typically maintain an action-value function ($Q$-function) and, optionally, a state-value function ($V$-function), referred to as $Q(s,a)$ and $V(s)$ respectively (Haarnoja et al., 2017; Nachum et al., 2017; Kumar et al., 2020; Kostrikov et al., 2021b). Define $Q^\pi : \mathcal{S} \times \mathcal{A} \to \mathbb{R}$, where $Q^\pi(s,a) = \mathbb{E}_\pi[\sum_{t=0}^\infty \gamma^t r(s_t, a_t)|s_0 = s, a_0 = a]$. The value function is learned by satisfying single-step Bellman consistencies. Let $\mathcal{T}^\pi$ be the Bellman operator with policy $\pi$ such that $(\mathcal{T}^\pi Q)(s,a) := r(s,a) + \gamma \mathbb{E}_{s'|s,a} \mathbb{E}_{a' \sim \pi}[Q(s', a')]$. Then $Q$ are learned by $\min_Q J(Q) = \frac{1}{2}\mathbb{E}_{(s,a)\sim\mathcal{D}}[(\mathcal{T}^\pi Q - Q)(s,a)^2]$.

**Behavior-regularized RL.**  In general, behavior-regularized RL considers the following constraint optimization problem with a reference distribution/policy $\mu$:

$$\pi^* = \arg\max_\pi \ \mathbb{E}_{s\sim\mathcal{D}, a\sim\pi(\cdot|s)}[R(s,a)] \ \text{ s.t. } \ \mathbb{E}_{s\sim\mathcal{D}}[M(\pi(\cdot|s), \mu(\cdot|s))] \leq \epsilon, \quad (1)$$

---

[1]Although VGF uses transport budget as an implicit regularization, here we use "without regularization" to emphasize that VGF doesn't involve any auxiliary regularization during optimization.

where $R(s, a)$ is a differentiable function and $M$ is some distance or divergence measure (e.g., KL-divergence, $L_2$ distance). $\mu$ could be the offline data distribution $\pi_\mathcal{D}$ in offline RL or the base model after pretraining in LLM RL. $R(s, a)$ could be either (multi-step) $Q$-function or (single-step) reward model in RL from human feedback (RLHF) (Bradley & Terry, 1952). We now briefly discuss several different approaches to Equation (1) and their limitations.

**(1) Policy gradient with $\pi$ reparameterized.** The most straightforward approach is to guide the policy to directly maximize function $R$ with reparameterized gradients. The constraint term will be added as a regularization term with a coefficient $\beta$ to balance these two gradients.

$$\max_\pi \mathbb{E}_{s \sim \mathcal{D}} \Big[ \mathbb{E}_{a \sim \pi} \big[ R(s, a) \big] - \beta \cdot M \big( \pi(\cdot|s), \mu(\cdot|s) \big) \Big]. \tag{2}$$

Reparameterized policy gradient is commonly used in offline RL with Gaussian policies (Wu et al., 2019; Fujimoto & Gu, 2021; Tarasov et al., 2024). However, extending this approach to large generative policies such as diffusion and flow matching models is challenging. These models generate actions through an iterative denoising process (Lipman et al., 2022). Reparameterized policy gradients require backpropagating through the sampling steps, which is often unstable and computationally expensive (Wang et al., 2023b). An alternative is to use distillation to compress the multi step policy into a one step model (Ding & Jin, 2023; Park et al., 2025b), but this will reduce expressivity.

**(2) Reject Sampling with $M = \mathrm{KL}$.** Using KL-divergence gives Equation (2) a closed-form solution which can be optimized by doing weighted behavior cloning (BC) (Peng et al., 2019; Xu et al., 2023) where actions are sampled from the reference policy, as follows.

$$\max_\pi \mathbb{E}_{s \sim \mathcal{D}, a \sim \mu} \big[ \exp(R(s, a)/\beta) \cdot \log \pi(a|s) \big]. \tag{3}$$

Although simple and easy to implement, using weighted BC tends to be mode-covering, only amplifying weak signals from the reference distribution without extracting new skills or knowledge (Wu et al., 2025). In fact, a simple best-of-$N$ sampling policy (Nakano et al., 2021), where $N$ i.i.d. samples are drawn from the reference distribution and one with the highest $R(s, a)$ is returned, is theoretically near optimal for this KL-constrained RL problem (Beirami et al., 2024; Yang et al., 2024).

$$\pi^* = \arg\max_{a_i \sim \mu, i \in [N]} R(s, a_i). \tag{4}$$

## 3 VALUE GRADIENT FLOW

VGF is designed to solve the above-mentioned challenges and provide a **simple and scalable** solution to Equation (1), and we give a detailed introduction in this section.

### 3.1 BEHAVIOR-REGULARIZED RL AS OPTIMAL TRANSPORT

We first consider a surrogate optimization objective that augments the value maximization objective in Equation (1) with a policy entropy maximization term: $\mathbb{E}_{a \sim \pi}[R(s, a)] + \alpha H(\pi(\cdot|s))$, where $H(\pi(\cdot|s)) \triangleq \mathbb{E}_\pi[-\log \pi(a|s)]$ is the causal entropy of the policy $\pi$ at state $s$. This Maximum-Entropy (MaxEnt) formulation of RL is well-known to enhance the exploration and robustness of the policy (Haarnoja et al., 2018; Garg et al., 2021; Eysenbach & Levine, 2021). However, our intuition here is that optimizing this MaxEnt objective turns the optimal policy distribution from greedy max to softmax over the whole action space, resulting a variational distribution as the Boltzmann distribution over the value function $R(s, a)$ (Ziebart, 2010; Bloem & Bambos, 2014):

$$\pi_R^*(a|s) = \frac{1}{Z_s} \exp\left(R(s, a)/\alpha\right), \tag{5}$$

where $Z_s$ is the normalization factor given as $\sum_{a'} \exp\left(R(s, a')/\alpha\right)$.

**Particle-based gradient flow.** We reframe the value-maximization problem as an optimal transport problem that transports probability mass from distribution $\mu$ to distribution $\pi_R^*$ defined in Equation (5). A natural way to formalize this transport is as a gradient flow of the functional $F(q) = \mathrm{KL}(q \,\|\, \pi_R^\star)$ on the space of probability measures endowed with the Wasserstein metric (Jordan et al., 1998; Ambrosio et al., 2008; Peyré & Cuturi, 2019). The resulting continuous-time evolution $q_t$ follows the continuity equation $\partial_t q_t + \nabla \cdot (q_t v_t) = 0$ with the steepest-descent velocity field $v_t = \nabla \log \pi_R^\star - \nabla \log q_t$,

so that $F(q_t)$ decreases monotonically and the stationary distribution is $\pi_R^\star$. However, directly solving $q_t$ is intractable, so we adopt the Jordan-Kinderlehrer-Otto (JKO) minimizing-movement scheme (Jordan et al., 1998) to obtain a discrete gradient flow:

$$q_{k+1} = \arg\min_q \ \mathrm{KL}(q \,\|\, \pi_R^\star) + \frac{1}{2h}\, W_2^2(q,\, q_k), \tag{6}$$

where $h > 0$ is the step size and $W_2$ is the 2-Wasserstein distance (Peyré & Cuturi, 2019). In Euclidean space, Equation (6) reduces to gradient descent on the function landscape. However, it is intractable as, in general, $q_k$ is infinite-dimensional.

To obtain a practical solver, we approximate $q_k$ by an empirical measure over $N$ particles in action space (for a fixed state $s$), $q_k \approx \frac{1}{N}\sum_{i=1}^N \delta_{a_i^{(k)}}$, and seek an update rule for $\{a_i^{(k)}\}_{i=1}^N$ that decreases Equation (6). By restricting the velocity field $v$ to the unit ball of a vector-valued reproducing kernel Hilbert space (RKHS), we get a solver that can be derived as the nonparametric functional gradient method that most rapidly decreases $\mathrm{KL}(q \,\|\, \pi_R^\star)$ within the RKHS (Liu & Wang, 2016; Liu, 2017). This yields a particle-based gradient flow solver that approximates the discrete gradient flow as $a_i^{(l+1)} = a_i^{(l)} + \epsilon \cdot \phi(a_i^{(l)})$, where

$$\phi(x) = \frac{1}{N}\sum_{j=1}^N \Big[ k(a_j,x) \underbrace{\nabla_{a_j}\log\pi_R^\star(a_j|s)}_{=\nabla_{a_j}R(s,a_j)/\alpha} + \nabla_{a_j}k(a_j,x) \Big] \xrightarrow{\text{w/o MaxEnt}} \frac{1}{N}\sum_{j=1}^N k(a_j,x)\nabla_{a_j}R(s,a_j). \tag{7}$$

Here, $\epsilon$ is the step size and $k(\cdot,\cdot)$ is the kernel function. The first term in $\phi(a_i)$ drives the particles toward the high probability regions of $\pi_R^*$ (i.e., with high $R(s,a)$), while the second term serves as a repulsive force to encourage dispersion and preserve multi-modality of the particles. The second term and $\alpha$ vanish when we transform the surrogate MaxEnt objective back to the original objective. This is equivalent to taking the limit $\alpha \to 0$ and absorbing $\alpha$ into the step size $\epsilon$.

Note that an **implicit behavior regularization** is imposed via controlling the transport budget ($L$, $\alpha$ and $\epsilon$). Intuitively, Equation (7) performs a kernel-smoothed transport to each particle in the action space, resulting in a controlled derivation from the reference distribution. Theoretically, we show in the following that the Maximum Mean Discrepancy (MMD) distance between the initial particles sampled from the reference policy and the particles generated by VGF is bounded.

**Theorem 1.** *Assume the value function $R(s,a)$ is c-Lipschitz w.r.t the input action $a$. Define the implicit policy that performs Equation (7) for $L$ steps with $N$ particles as $\pi_N^L$. We have*

$$\mathrm{MMD}^2(\mu, \pi_N^L) = \mathrm{MMD}^2(\pi_N^0, \pi_N^L) \le \frac{2\epsilon L}{\sigma\sqrt{e}}\left(\frac{c}{\alpha} + \frac{1}{\sigma\sqrt{e}}\right).$$

**VGF in the LLM setting.** In the LLM RL setting, at time step $t$, the action $a_t$ is a discrete token and the state $s_t$ is the token sequence $s_t = (x_0,\ldots,x_L,a_0,\ldots,a_{t-1})$, where $x = (x_0,\ldots,x_L)$ is the input prompt and $y = (a_0,\ldots,a_{t-1})$ are the generated tokens up to step $t-1$. The transition function $P$ updates the state deterministically via concatenation: $s_{t+1} = P(s_t,a_t) = s_t \,\|\, a_t$.

A direct application of Eq. (7) to tokens is infeasible because tokens are discrete. We therefore perform VGF in a **continuous surrogate space** and decode back to the discrete token space only at the end of the gradient flow. Let $u$ be a differentiable representation of a full response $y$. The representation could either be the token-embedding matrix $u \in \mathbb{R}^{T\times d}$ or a latent vector $u = z \in \mathbb{R}^m$ of a flow or diffusion language model with $y = \mathrm{Dec}(z)$. Denote $y_i^{(l)} = \mathrm{Dec}(u_i^{(l)})$. Because the reward model is differentiable with respect to its input embeddings, response-level gradient $\nabla_y R(x,y)$ can be back-propagated to the surrogate via the chain rule as follows.

$$\nabla_{u_i}\log\pi_R^*\big(y_i^{(l)}|x\big) = \frac{1}{\alpha}\, J_i^\top\, \nabla_y R\big(x, y_i^{(l)}\big), \quad J_i := \frac{\partial \mathrm{Dec}\big(u_i^{(l)}\big)}{\partial u_i^{(l)}}. \tag{8}$$

One motivation to use VGF is that the SFT policy is far from random, with most probability mass concentrating on a small subset of tokens and modes. Note that VGF utilizes first-order gradient guidance from $R$, this avoids high-variance PPO-style optimization (Ouyang et al., 2022) and enables

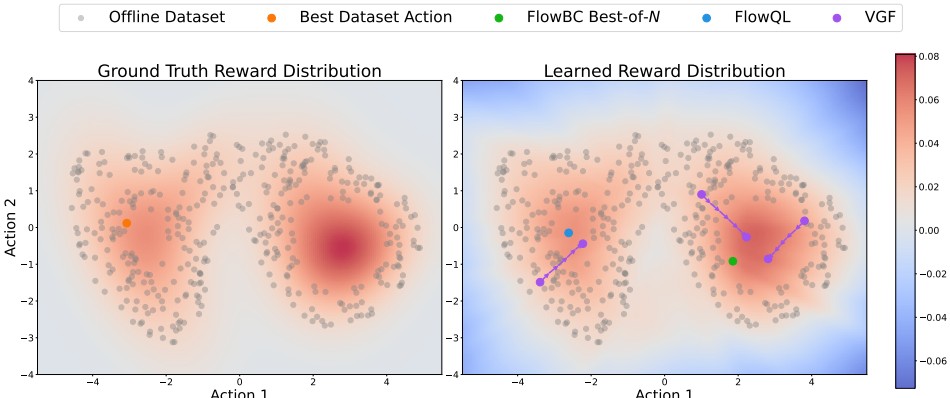

Figure 2: **Toycase results**. VGF generates actions with higher ground-truth reward than other methods.

inference-only control similar to best-of-$N$ sampling. However, the difference is that VGF steers particles toward high-reward modes, the resulting implicit policy need not remain within the support of the reference distribution, as shown in the following theorem.

**Theorem 2.** *Define the $\epsilon$-support of a distribution $P$ as $\mathrm{supp}_\epsilon(P) := \{x : p(x) \geq \epsilon\}$. We have*

$$\mathrm{supp}_\epsilon(\pi_N^L(\cdot|s)) \not\subseteq \mathrm{supp}_\epsilon(\mu(\cdot|s)).$$

This indicates that, unlike the methods discussed in Section 2 which result in $\mathrm{supp}_\epsilon(\pi(\cdot|s)) \subseteq \mathrm{supp}_\epsilon(\mu(\cdot|s))$ (Wu et al., 2025), VGF breaks the over-conservative behavior constraint, enabling the discovery and exploitation of novel behaviors beyond the reference distribution.

## 3.2 DISCUSSION

Below, we outline several distinctive advantages of VGF and discuss the connection with prior work.

**(1) Stable optimization with reduced conservatism.** Compared with methods based on reparameterized policy gradient (Wang et al., 2023a), VGF avoids optimization instability caused by backpropagating through time. Futhermore, VGF is optimized to find the best reward-maximization policy within a fixed behavior constraint, which is more aligned with Equation (1).

**(2) Implicit policy with multimodal expressivity.** While bypassing explicit policy parameterization, the implicit policy in VGF is still expressive enough to capture a multimodal distribution. Owing to the usage of gradient flow, VGF naturally preserves and sharpens multiple high-value modes from the reference distribution instead of collapsing to a single one. This is different from BCQ (Fujimoto et al., 2019) where a Gaussian residual policy with limited expressivity is learned on top of the reference policy. Note that in the offline RL setting, VGF remains versatile to the usage of different advanced generative models, e.g., Diffusion models (Song et al., 2021) or Flow models (Lipman et al., 2022), to generate samples from the reference distribution given only an offline dataset.

**(3) Adaptive scaling during test-time.** One intriguing property of VGF is that it enables adaptive test-time scaling via varying the transport budget **without** any retraining. For example, when the value function $R$ can generalize well, the performance will scale with the number of test-time flow steps, which could be different from the number of train-time flow steps. However, when the value function has large extrapolation errors, by setting the test-time flow steps to 0, VGF reduces to Best-of-$N$ sampling methods (Chen et al., 2023; Hansen-Estruch et al., 2023). One difference in this case is that VGF learns the value function by TD learning (since the train-time flow step is not 0) instead of in-sample learning (Kostrikov et al., 2021b; Xu et al., 2023). We find in practice that TD learning enables better stitching and generalization. This difference makes VGF **fundamentally different** from Diffusion-based methods (Mao et al., 2024; Frans et al., 2025) that can also do adaptive generation via adjusting the guidance weight but rely on in-sample value learning.

**Practical consideration.** We summarize the pseudo-code in Algorithm 1. To reduce sampling cost, we use a small number of VGF particles ($N = 5$) across all experiments. At test time, since we need to choose one particle to do evaluation, we use best-of-$N$ sampling from all VGF particles based on the value/reward function. In offline RL, given the offline dataset, we train a behavior cloning policy

to generate samples from $\mu$. The $Q$-function is trained using TD-learning, and we average over all particles when computing the target $Q$-values. We use the **w/o MaxEnt** objective in Equation (7). To accelerate training and inference, we additionally train a network $f(s,a)$ to capture the gradient of $Q$-function via $\min_f \mathbb{E}_{(s,a)\sim\mathcal{D}}[(f(s,a) - \nabla_a Q(s,a))^2]$.

**A toy example.** We use a toy example to illustrate the mechanism of VGF. We construct a 2-D continuous control bandit task with a bimodal ground-truth reward distribution, where the offline dataset is generated from sampling from sub-optimal reward regions, as demonstrated in Figure 2. We are interested in studying the behavior of the following three different behavior-regularized RL methods. Note that all three methods fit a learned reward model using $L_2$ loss and a BC flow model using flow-matching loss. **FlowQL** (Park et al., 2025b): This method represents the first group of methods in section 2. FlowQL additionally trains a one-step flow model as the policy, which is used during evaluation. We carefully tune the coefficient $\beta$ to ensure the best performance. **FlowBC Best-of-$N$**: This method represents the second group of methods in section 2. In this case, we sample $N = 20$ actions from the BC flow model and do best-of-$N$ sampling using the learned reward function. **VGF**: For this task, we set particle number $N = 3$, $L_{\text{test}} = 5$ and $\alpha = 0.1$.

---

**Algorithm 1** Value Gradient Flow

1: **function** VGF$(s, \hat{\mu}, R, L_{\text{test}})$
2:     Get $a_N^0 \sim \hat{\mu}(\cdot|s)$
3:     **for** $l = 0, 1, \ldots, L_{\text{test}} - 1$ **do**
4:       Get $a_N^{l+1}$ using $R$ and $a_N^l$ by Eq. (7)
5:     **return** $a_N^{L_{\text{test}}}$

**Require:** $\mathcal{D}, L_{\text{train}}, L_{\text{test}}, \epsilon$
6:  ▷ Value Training (offline RL)
7: **for** $t = 1, 2, \cdots, M$ **do**
8:     Sample transitions $(s, a, r, s') \sim \mathcal{D}$
9:     Train behavior cloning policy $\hat{\mu}$
10:     Get $a_N^{L_{\text{train}}} = \text{VGF}(s', \hat{\mu}, Q, L_{\text{train}})$
11:     Train $Q$ using $a_N^{L_{\text{train}}}$ by TD learning
12: ▷ Evaluation (offline RL and RLHF)
13: Get initial state $s$, set $d$ as False
14: **while** not $d$ **do**
15:     Get $a_N^{L_{\text{test}}} = \text{VGF}(s, \hat{\mu}, R, L_{\text{test}})$
16:     Get best-of-$N$ $a^*$ from $a_N^{L_{\text{test}}}$ by Eq. (4)
17:     Roll out $a^*$ and get $(s', r, d)$
18:     Set $s \leftarrow s'$

---

We plot the action generated by FlowQL and FlowBC Best-of-$N$, along with the flow trajectories of particles generated by VGF, in Figure 2. As shown, even if the learned reward model gets some error, the implicit policy in VGF demonstrates successful and effective exploration of the area with high ground-truth reward. By contrast, FlowQL is shown to be misled by the error of the learned reward model, generating actions with suboptimal values. Although best-of-$N$ sampling from the FlowBC model could improve the best dataset action, it still falls within the support of the suboptimal behavior distribution due to over-conservatism, which aligns well with the theoretical analysis.

## 4   RELATED WORK

**Offline RL.** To address distributional shift, most model-free offline RL methods augment off-policy learning with behavior regularization. This can appear explicitly as divergence penalties that constrain the learned policy toward the dataset distribution (Wu et al., 2019; Kumar et al., 2019; Fujimoto & Gu, 2021) or implicitly via weighted behavior cloning and advantage-weighted updates (Wang et al., 2020; Nair et al., 2020). To improve policy expressiveness, recent work adopts expressive generative models as policies, including Decision Transformer (Chen et al., 2021), diffusion-based policies (Wang et al., 2023b; Chen et al., 2023; Hansen-Estruch et al., 2023), flow matching policies (Park et al., 2025b) and consistency-model policies (Ding & Jin, 2023). Compared with these methods, VGF bypasses explicit policy parameterization, improving training stability while retaining strong expressiveness. There are several recent works that use the $Q$ action gradient to guide the policy learning (Yang et al., 2023; Psenka et al., 2023; Mark et al., 2024; Xu et al., 2025b; Li & Levine, 2026). However, their motivations and derivations are fundamentally different from VGF. Among them, the most closely related methods are PA-RL (Mark et al., 2024) and QAM (Li & Levine, 2026). Compared to PA-RL, VGF avoids explicitly parameterizing a policy, which can otherwise limit flexibility for adaptive test-time scaling. In contrast to QAM, which optimizes a KL-regularized policy that still remains within the support of the behavior distribution, VGF is encouraged to move beyond the behavior distribution due to its implicit regularization.

**Reinforcement Learning from Human Feedback.** In RLHF, policy models can exploit imperfections in learned reward models, a phenomenon often termed reward over-optimization (Gao et al., 2023) and also discussed as reward hacking or reward gaming (Amodei et al., 2016; Skalse et al., 2022; Pang et al., 2023). Many studies analyze this effect in synthetic setups that substitute expensive

Table 1: **D4RL offline RL results**. Scores are averaged over the final 10 evaluations across 5 seeds with standard deviation reported, we highlight the best score in integer-level. VGF demonstrates superior performance on most tasks, especially those challenging ones.

| Dataset | Gaussian Policy | | | Diffusion/Flow Policy | | | w/o Policy |
| --- | --- | --- | --- | --- | --- | --- | --- |
| | TD3+BC | IQL | IVR | Diffusion-QL | SfBC | FQL | VGF (ours) |
| halfcheetah-m | 48.3 | 47.4 $\pm0.2$ | 48.3 $\pm0.2$ | 51.1 $\pm0.5$ | 45.9 $\pm2.2$ | 55.6 $\pm0.2$ | 57.1 $\pm0.1$ |
| hopper-m | 59.3 | 66.3 $\pm5.7$ | 75.5 $\pm3.4$ | 90.5 $\pm4.6$ | 57.1 $\pm4.1$ | 60.6 $\pm0.1$ | 97.9 $\pm2.0$ |
| walker2d-m | 83.7 | 72.5 $\pm8.7$ | 84.2 $\pm4.6$ | 87.0 $\pm0.9$ | 77.9 $\pm2.5$ | 65.9 $\pm0.3$ | 89.4 $\pm0.8$ |
| halfcheetah-m-r | 44.6 | 44.2 $\pm1.2$ | 44.8 $\pm0.7$ | 47.8 $\pm0.3$ | 37.1 $\pm1.7$ | 48.3 $\pm0.3$ | 49.1 $\pm0.1$ |
| hopper-m-r | 60.9 | 95.2 $\pm8.6$ | 99.7 $\pm3.3$ | 101.3 $\pm0.6$ | 86.2 $\pm9.1$ | 50.7 $\pm2.7$ | 99.0 $\pm1.1$ |
| walker2d-m-r | 81.8 | 76.1 $\pm7.3$ | 81.2 $\pm3.8$ | 95.5 $\pm1.5$ | 65.1 $\pm5.6$ | 38.8 $\pm1.1$ | 97.8 $\pm1.6$ |
| halfcheetah-m-e | 90.7 | 86.7 $\pm5.3$ | 94.0 $\pm0.4$ | 96.8 $\pm0.3$ | 92.6 $\pm0.5$ | 102.1 $\pm0.6$ | 99.1 $\pm0.3$ |
| hopper-m-e | 98.0 | 101.5 $\pm7.3$ | 111.8 $\pm2.2$ | 111.1 $\pm1.3$ | 108.6 $\pm2.1$ | 76.7 $\pm0.6$ | 98.3 $\pm3.3$ |
| walker2d-m-e | 110.1 | 110.6 $\pm1.0$ | 110.0 $\pm0.8$ | 110.1 $\pm0.3$ | 109.8 $\pm0.2$ | 102.6 $\pm0.2$ | 110.5 $\pm1.5$ |
| antmaze-u | 78.6 | 85.5 $\pm1.9$ | 92.2 $\pm1.4$ | 93.4 $\pm3.4$ | 92.0 $\pm2.1$ | 96 $\pm1.6$ | 98.0 $\pm1.8$ |
| antmaze-u-d | 71.4 | 66.7 $\pm4.0$ | 74.0 $\pm2.3$ | 66.2 $\pm8.6$ | 85.3 $\pm3.6$ | 89 $\pm2.3$ | 94.3 $\pm1.4$ |
| antmaze-m-p | 10.6 | 72.2 $\pm5.3$ | 80.2 $\pm3.7$ | 76.6 $\pm10.8$ | 81.3 $\pm2.6$ | 78.0 $\pm2.6$ | 89.4 $\pm3.1$ |
| antmaze-m-d | 3.0 | 71.0 $\pm3.2$ | 79.1 $\pm4.2$ | 78.6 $\pm10.3$ | 82.0 $\pm3.1$ | 71.0 $\pm3.4$ | 86.7 $\pm2.8$ |
| antmaze-l-p | 0.2 | 39.6 $\pm4.5$ | 53.2 $\pm4.8$ | 46.4 $\pm8.3$ | 59.3 $\pm14.3$ | 84.0 $\pm2.9$ | 82.5 $\pm3.6$ |
| antmaze-l-d | 0.0 | 47.5 $\pm4.4$ | 52.3 $\pm5.2$ | 56.6 $\pm7.6$ | 45.5 $\pm6.6$ | 83.0 $\pm3.8$ | 83.8 $\pm4.5$ |

human evaluation with strong "gold" models for labeling and assessment (Gao et al., 2023; Moskovitz et al., 2024; Coste et al., 2023). A prevailing mitigation strategy augments the reward or training objective with a KL penalty to a supervised-finetuned reference model (Kullback & Leibler, 1951; Stiennon et al., 2020; Ouyang et al., 2022; Bai et al., 2022). Other approaches employ ensembles or early-stopping-style constraints to curb over-optimization while controlling KL (Coste et al., 2023; Moskovitz et al., 2024). However, explicit penalties inevitably introduce a reward-KL trade-off that is sensitive to coefficient tuning and can be overly conservative towards the reference support.

**Optimal transport in RL.** Optimal transport (OT) provides a geometry over distributions that has proved useful in multiple RL settings. In distributional RL, Wasserstein metrics underpin return-distribution learning via quantile-regression objectives, improving stability and control (Dabney et al., 2018). OT has also been used to align occupancy distributions for imitation and offline learning. For example, Sinkhorn-based matching or primal Wasserstein formulations that shape rewards and enable cross-domain alignment (Dadashi et al., 2021). Beyond matching, an OT viewpoint motivates transporting probability mass toward value-preferred regions, inspiring flow/particle-style policy updates and robust formulations that explicitly constrain distributional shift. Our work follows this trajectory but emphasizes transport from the reference distribution to the optimal policy distribution using value gradients. Other work like PPL (Asadulaev et al., 2024) considers transport between states and partial action distributions, whereas VGF operates in the action space.

## 5 EXPERIMENTS

The goal of our experiments is to evaluate the efficacy of VGF in improving offline RL and RLHF. We evaluate the performance of VGF on D4RL and OGBench and compare it with prior methods. We also provide an ablation study on important hyperparameters and investigate the adaptive scaling behavior during test time in VGF to gain a deeper understanding of its mechanism.

### 5.1 OFFLINE RL RESULTS

**D4RL Benchmark Datasets.** We evaluate the performance of VGF on the D4RL benchmark (Fu et al., 2020), and compare it with several algorithms based on Gaussian policy, diffusion policy and flow policy. Gaussian-policy-based baselines include TD3+BC (Fujimoto & Gu, 2021), IQL (Kostrikov et al., 2021a), and IVR (Xu et al., 2023). We also select Diffusion-QL (Wang et al., 2023b) and SfBC (Chen et al., 2023) as diffusion-policy-based baselines, and FQL (Park et al., 2025b) as a flow-policy-based baseline. The evaluation tasks include MuJoCo, a set of locomotion tasks, and AntMaze, a series of navigation tasks. The results in Table 1 show that VGF outperforms all of the baselines in most datasets. Note that FQL has poor performance on some Mujoco datasets even

Table 2: **OGBench offline RL results**. Scores are averaged over the final 10 evaluations across 5 seeds with standard deviation reported, we highlight the best score in integer-level. VGF achieves competitive or superior performance compared to prior approaches, especially on hard tasks.

| Dataset (5 tasks) | Gaussian Policy | | | Diffusion/Flow Policy | | | w/o Policy |
|---|---|---|---|---|---|---|---|
| | BC | IQL | ReBRAC | FBRAC | IDQL | FQL | VGF (ours) |
| antmaze-giant | $0 \pm 0$ | $4 \pm 1$ | $26 \pm 8$ | $4 \pm 4$ | $0 \pm 0$ | $9 \pm 6$ | $3 \pm 1$ |
| humanoidmaze-medium | $2 \pm 1$ | $33 \pm 2$ | $22 \pm 8$ | $38 \pm 5$ | $1 \pm 0$ | $58 \pm 5$ | $72 \pm 1$ |
| humanoidmaze-large | $1 \pm 0$ | $2 \pm 1$ | $2 \pm 1$ | $2 \pm 0$ | $1 \pm 0$ | $4 \pm 2$ | $15 \pm 2$ |
| antsoccer-arena | $1 \pm 0$ | $8 \pm 2$ | $0 \pm 0$ | $16 \pm 1$ | $12 \pm 4$ | $60 \pm 2$ | $63 \pm 4$ |
| cube-single | $5 \pm 1$ | $83 \pm 3$ | $91 \pm 2$ | $79 \pm 7$ | $95 \pm 2$ | $96 \pm 1$ | $96 \pm 1$ |
| cube-double | $2 \pm 1$ | $7 \pm 1$ | $12 \pm 1$ | $15 \pm 3$ | $15 \pm 6$ | $29 \pm 2$ | $70 \pm 8$ |
| scene | $5 \pm 1$ | $28 \pm 1$ | $41 \pm 3$ | $45 \pm 5$ | $46 \pm 3$ | $56 \pm 2$ | $60 \pm 1$ |
| puzzle-3x3 | $2 \pm 0$ | $9 \pm 1$ | $21 \pm 1$ | $14 \pm 4$ | $10 \pm 2$ | $30 \pm 1$ | $75 \pm 4$ |
| puzzle-4x4 | $0 \pm 0$ | $7 \pm 1$ | $14 \pm 1$ | $13 \pm 1$ | $29 \pm 3$ | $17 \pm 2$ | $45 \pm 4$ |

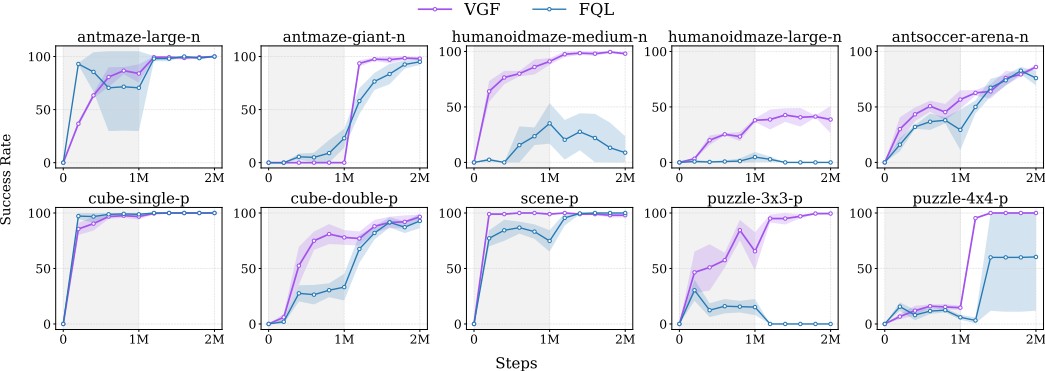

Figure 3: **OGBench offline-to-online RL results.** Learning curves for online fine-tuning of VGF and FQL across all default tasks. VGF not only provides a stronger initialization from offline training but also leads to faster adaptation and higher final success rates. The shaded gray area denotes offline training.

under carefully hyperparameter tuning. We would like to highlight that VGF achieves much higher scores than prior methods on challenging AntMaze datasets.

**OGBench Datasets.** Another benchmark that we use to evaluate is OGBench (Park et al., 2025a), which provides a variety of goal-conditioned tasks and datasets across robotic locomotion and manipulation. Among the reward-based single-task settings, we select 4 locomotion and 5 manipulation environments, each of which consists of 5 tasks. We take the average score of the 5 tasks to be the indicator of performance in these environments. We compare our results with several of the state-of-the-art algorithms, including ReBRAC (Tarasov et al., 2023), Flow BRAC (Wu et al., 2019), IDQL (Hansen-Estruch et al., 2023) and FQL (Park et al., 2025b). ReBRAC leverages a monolithic Q network and a Gaussian policy and achieves competitive performance on offline RL datasets. Flow BRAC replaces Gaussian policy with flow policy in behavior-regularized actor-critic algorithms. IDQL trains a diffusion BC model along with a value function learned by IQL to do best-of-$N$ sampling. FQL is a recently proposed method that introduces a one-step flow policy that distills from the BC flow policy to avoid unstable gradient backpropagation through time. Table 2 summarizes the results of VGF and its comparison with baselines on OGBench. VGF achieves better performance than prior methods in most of the environments, especially those hard ones where FQL attains performance below 50% success rate (`cube-double`, `puzzle-3x3` and `puzzle-4x4`).

**Online Finetuning.** We evaluate the efficacy of VGF in the online RL fine-tuning setting. Here, we first train agents offline for 1M steps and subsequently fine-tune them online for an additional 1M steps. Across all challenging tasks, we find that VGF provides a stronger initialization compared to FQL and also adapts more rapidly during online interaction and converges to higher final performance.

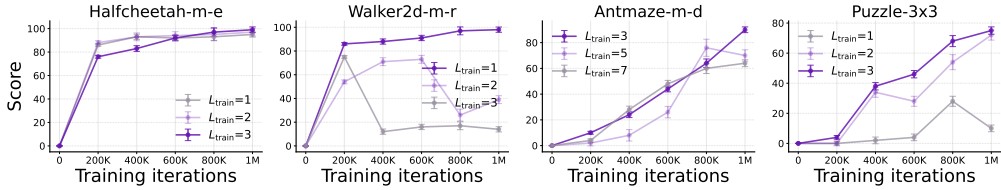

Figure 4: Ablation study on VGF train-time flow steps $L_{\text{train}}$.

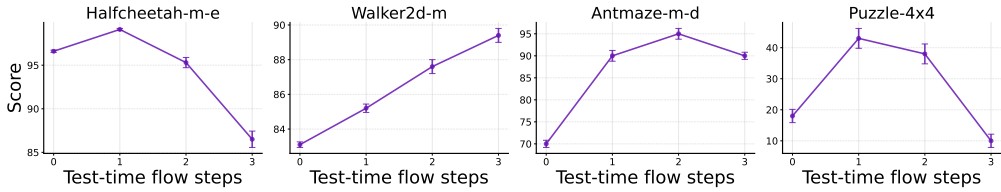

Figure 5: VGF enables adaptive test-time scaling behavior by adjusting test-time flow steps $L_{\text{test}}$.

## 5.2 RLHF RESULTS

We report results on the TL;DR Summarize (Stiennon et al., 2020) and Anthropic Helpful and Harmless Dialogue (Bai et al., 2022) datasets. The training split of `TL;DR` dataset contains 116k human-written instructions and 93k human-annotated preference pairs. The preprocessed `Anthropic-HH` dataset contains 112k training preference pairs. Both the reward model and the SFT policy are initialized from the same Pythia-2.8B base model, trained on either human demonstrations (`TL;DR`) or the chosen responses (`Anthropic-HH`). For evaluation metrics, we calculate win rates (WR%) judged by GPT-4 comparing outputs from the initialization and aligned models. As shown in Table 3, VGF outperforms all baseline RLHF methods by a large margin.

| Model | TL;DR WR% (vs ref) | Anthropic-HH WR% (vs chosen) |
|---|---|---|
| Pythia-SFT | 48.5 | 46.2 |
| PPO | 57.3 | 45.5 |
| DPO | 61.2 | 51.5 |
| Best-of-$N$ | 58.3 | 49.0 |
| VGF (Ours) | 68.1 | 59.0 |

Table 3: **RLHF results**. VGF outperforms baseline RLHF methods by having higher win-rates on TL;DR and Anthropic-HH dataset.

## 5.3 UNDERSTANDING THE PROPERTIES AND BEHAVIOR OF VGF

To better understand the mechanism and behavior of VGF, we investigate the importance of hyperparameter selection and the test-time scaling property by conducting some ablation study.

**What are the important hyperparameters of VGF?** There are three hyperparameters in VGF: train-time flow steps $L_{\text{train}}$, step size $\epsilon$ and particle number $N$, with $L_{\text{train}}$ being the most important one. Train-time flow steps $L_{\text{train}}$ is the number of flow steps we adopt during training. $L$ is directly related to the distance between the reference policy and the learned policy. Intuitively, a larger $L$ means deviating more from the reference policy. In Figure 4, we show that optimal $L_{\text{train}}$ needs to be tuned per task to achieve the best performance.

**Can VGF enable adaptive test-time scaling behavior?** The answer is yes by controlling the test-time flow steps $L_{\text{test}}$. However, the optimal $L_{\text{test}}$ depends on the specific dataset. In general, scaling up the test-time flow steps is helpful when the value function generalizes well to OOD regions and the offline dataset is of low quality, which requires improving over the reference policy to achieve the best performance. Note that even when the value function has large extrapolation error, by setting the test-time flow steps to 0, VGF reduces to the best-of-N sampling method, but it can still outperform the reference policy, enabling in-distribution generalization owing to the training of a value function via TD learning.

## 6 LIMITATIONS AND FUTURE WORK

In this paper, we propose VGF, a new scalable paradigm that casts behavior-regularized RL as optimal transport from the reference distribution to the optimal policy distribution induced by the value function, where the transport budget serves as the implicit behavior regularization. VGF uses particle-based gradient flow as the practical solution, yielding an implicit policy with multimodal expressivity while bypassing the need of policy reparameterization. VGF is easy to implement, enabling adaptive test-time scaling and achieves strong empirical results on both offline RL and RLHF tasks. We believe that VGF represents a concrete step toward building unified and scalable behavior-regularized RL algorithms. One limitation of VGF lies in its ability to handle scenarios where the reference distribution is heavily skewed toward suboptimal behavior. Using technique like distribution reweighting (Xu et al., 2025a) to enhance performance is one future work. Another promising direction is to integrate VGF with methods that further improve the expressiveness of the value function (Agrawalla et al., 2025; Dong et al., 2025; 2026), with the goal of boosting performance on long-horizon tasks.

## 7 REPRODUCIBILITY STATEMENT

To ensure the reproducibility of this paper, we detail the theoretical and empirical parts of our results in the main paper and the appendix. In Section 3, we introduce the basic notations used in the theoretical analysis and establish theories to support our claim. Furthermore, in Appendix A, we provide the detailed proofs of the theoretical results. For empirical details, we briefly introduce the setup of our toy case in Section 3. In Appendix B, we elaborate on the benchmark environments, the network architecture and hyperparameters in our experiments, and provide a more detailed version of empirical results on OGBench. In Appendix C, we provide a simple implementation of our method. We will release the code after acceptance.

## ACKNOWLEDGEMENT

We thank members from MIDI lab for discussions on the method and feedback on the early draft of the paper. This work is partially supported by NSF 2340651, NSF 2402650, DARPA HR00112490431, ARO W911NF-24-1-0193, U. S. Army Research Laboratory and the U. S. Army Research Office under Grant W911NF2010219 and ONR N00014-26-1-2055. HX is supported by the Amazon Fellowship. This research used the computational cluster resource provided by the Texas Advanced Computing Center at UT Austin and Jetstream2 at Indiana University through allocation CIS250850 from the Advanced Cyberinfrastructure Coordination Ecosystem: Services & Support (ACCESS) program, which is supported by National Science Foundation grants #2138259, #2138286, #2138307, #2137603, and #2138296.

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

## A    PROOF

### A.1    PROOF OF THEOREM 1

**Theorem 1.** Assume the value function $R(s, a)$ is $c$-Lipschitz w.r.t the input action $a$. Define the implicit policy that performs Equation (7) for $L$ steps with $N$ particles as $\pi_N^L$. We have

$$\text{MMD}^2(\mu, \pi_N^L) = \text{MMD}^2(\pi_N^0, \pi_N^L) \leq \frac{2\epsilon L}{\sigma\sqrt{e}} \left( \frac{c}{\alpha} + \frac{1}{\sigma\sqrt{e}} \right).$$

*Proof.* When $R(s, a)$ is $c$-Lipschitz, we know that $\|\nabla_{a_j} R(s, a_j)\|_\infty \leq c$. Additionally, we have another key observation that the kernel $k$ also has Lipschitz property. Formally,

$$k(x + \delta, y) - k(x, y) = \exp(-\|x - y\|^2/2\sigma^2) - \exp(-\|x + \delta - y\|^2/2\sigma^2) \leq \frac{\|\delta\|_\infty}{\sigma\sqrt{e}}.$$

In other words, the kernel $k$ is $\frac{1}{\sigma\sqrt{e}}$-Lipschitz. Combining these two properties, we have

$$\|h(s, x)\|_\infty = \|\mathbb{E}_{a_j \sim \mu_0} \left[ k(a_j, x)\nabla_{a_j} R(s, a_j)/\alpha + \nabla_{a_j} k(a_j, x) \right] \|_\infty$$
$$\leq \sup_{a_j} k(a_j, x) \cdot \frac{c}{\alpha} + \sup_{a_j} \|\nabla_{a_j} k(a_j, x)\|_\infty$$
$$\leq \frac{c}{\alpha} + \frac{1}{\sigma\sqrt{e}}.$$

The square of MMD distance is calculated as follows:

$$\text{MMD}^2(\pi_N^L, \pi_N^0) = \mathbb{E}_{x,y \sim \pi_N^0} k(x, y) + \mathbb{E}_{x,y \sim \pi_N^L} k(x, y) - 2\mathbb{E}_{x \sim \pi_N^0, y \sim \pi_N^L} k(x, y).$$

Define $K := \frac{1}{\sigma\sqrt{e}}$ and $H := \frac{c}{\alpha} + \frac{1}{\sigma\sqrt{e}}$. Let us consider the condition of a single pair of particles $x, y$ after one iteration. We have

$$k(x, y) + k(x + \epsilon h(x), y + \epsilon h(y)) - k(x + \epsilon h(x), y) - k(x, y + \epsilon h(y))$$
$$\leq |k(x, y) - k(x + \epsilon h(x), y)| + |k(x + \epsilon h(x), y + \epsilon h(y)) - k(x, y + \epsilon h(y))|$$
$$\leq 2\epsilon K H.$$

Denote $x^{(k)}$ as the particle $x$ after the $k$-th iteration, we have

$$\text{MMD}^2(\pi_N^L, \pi_N^0) = \mathbb{E}_{x,y \sim \pi_N^0} k(x, y) + \mathbb{E}_{x,y \sim \pi_N^L} k(x, y) - 2\mathbb{E}_{x \sim \pi_N^0, y \sim \pi_N^L} k(x, y)$$
$$= \frac{1}{n^2} \sum_{x,y} \left[ k(x^{(0)}, y^{(0)}) + k(x^{(L)}, y^{(L)}) - k(x^{(L)}, y^{(0)}) - k(x^{(0)}, y^{(L)}) \right]$$
$$\leq 2\epsilon K H L.$$

$\square$

### A.2    PROOF OF THEOREM 2

**Theorem 2.** Define the $\epsilon$-support of a distribution $P$ as $\text{supp}_\epsilon(P) := \{x : p(x) \geq \epsilon\}$. We have

$$\text{supp}_\epsilon(\pi_N^L(\cdot|s)) \not\subseteq \text{supp}_\epsilon(\mu(\cdot|s)).$$

The proof can be divided into two settings, where the policy distribution is discrete and continuous. We first present the simple proof for the discrete setting.

*Proof.* Denote the support of the behavioral policy as $\text{supp}(\pi_N^0) = \{a_1, ..., a_N\}$. If the support of the learned policy from one-step SVGD $\text{supp}(\pi_N^1)$ is a subset of $\text{supp}(\pi_N^0)$, then we know that $a_1$ is updated to $a_i \in \text{supp}(\pi_N^0)$. This indicates that

$$a_i = a_1 + \epsilon h(a_1) = a_1 + \epsilon \mathbb{E}_{a_j \sim \pi_N^0} \left[ k(a_j, a_1)\nabla_{a_j} R(s, a_j)/\alpha + \nabla_{a_j} k(a_j, a_1) \right].$$

Note that a little disturbance in any of the dimensions of $\nabla_{a_j} R(s, a_j)$ will make the equation invalid when the other derivatives of $R$ are fixed. Therefore, the equation is almost surely invalid.    $\square$

As for the continuous setting, we typically assume that the policies before and after an SVGD update both follow Gaussian distributions, which is a widely used assumption. Formally, we assume that $\pi_N^0 \sim \mathcal{N}(\mu_1, \sigma_1^2), \pi_N^1 \sim \mathcal{N}(\mu_2, \sigma_2^2)$.

*Proof.* We consider the gradient of an arbitrary particle $a$ sampled from $\pi_N^0$. We know that

$$h(a) = \mathbb{E}_{a_j \sim \pi_N^0} \left[ k(a_j, a) \nabla_{a_j} R(s, a_j)/\alpha + \nabla_{a_j} k(a_j, a) \right].$$

Let us investigate the first term. We also know that

$$\begin{aligned}
\nabla_{a_j} R(s, a_j) &= \nabla_{a_j} \log \pi_N^1(a_j|s) \\
&= \nabla_{a_j} \log \left[ \frac{1}{\sqrt{2\pi}\sigma_2} \exp(-\frac{(a_j - \mu_2)^2}{2\sigma_2^2}) \right] \\
&= \frac{-\frac{1}{\sqrt{2\pi}\sigma_2} \exp(-\frac{(a_j - \mu_2)^2}{2\sigma_2^2}) \frac{a_j - \mu_2}{\sigma_2^2}}{\frac{1}{\sqrt{2\pi}\sigma_2} \exp(-\frac{(a_j - \mu_2)^2}{2\sigma_2^2})} \\
&= \frac{\mu_2 - a_j}{\sigma_2^2}.
\end{aligned}$$

Then, we calculate the second term:

$$\nabla_{a_j} k(a_j, a) = k(a_j, a) \frac{a_j - a}{\sigma_1^2}.$$

Combining both terms, we have

$$\begin{aligned}
h(a) &= \mathbb{E}_{a_j \sim \pi_N^0} \left[ k(a_j, a)(\frac{\mu_2 - a_j}{\alpha\sigma_2^2} + \frac{a_j - a}{\sigma_1^2}) \right] \\
&= \mathbb{E}_{a_j \sim \pi_N^0} \left[ k(a_j, a) \frac{\mu_1 - a_j}{\alpha\sigma_2^2} \right] + \mathbb{E}_{a_j \sim \pi_N^0} \left[ k(a_j, a)(\frac{\mu_2 - \mu_1}{\alpha\sigma_2^2} + \frac{a_j - a}{\sigma_1^2}) \right] \\
&= \mathbb{E}_{a_j \sim \pi_N^0} \left[ k(a_j, a)(\frac{\mu_2 - \mu_1}{\alpha\sigma_2^2} + \frac{a_j - a}{\sigma_1^2}) \right].
\end{aligned}$$

The last equation above is because of the symmetry of Gaussian distributions. We now focus on the first dimension of particles. Without loss of generality, we assume that $\mu_{1,1} \leq \mu_{2,1}$. From the $\epsilon$-support of $\pi_N^0$, which is a closed region, we choose $a$ with the smallest value in the first dimension, i.e., $(a - x)_1 \leq 0$ for any $x \in \text{supp}(\pi_N^0)$. This indicates that the first dimension of $h(a)$ is strictly greater than zero, which means that the updated particle of $a$ is out of the $\epsilon$-support of $\pi_N^0$. $\square$

## B  EXPERIMENTAL DETAILS

### B.1  OFFLINE RL EVALUATION DETAILS

**Environments, tasks, and datasets.** In the offline setting, VGF is evaluated on different kinds of datasets from various environments.

For MuJoCo environments, we have the following datasets.

- `halfcheetah/hopper/walker2d-m` (medium): Collected by a policy with moderate performance, typically reaching around one-third of expert returns. These datasets represent structured but suboptimal behavior.

- `halfcheetah/hopper/walker2d-m-r` (medium-replay): Contains the replay buffer of the mediocre SAC policy. It includes a wide range of off-policy transitions, many of which are suboptimal or noisy.

- `halfcheetah/hopper/walker2d-m-e` (medium-expert): A 50-50 mixture of medium and expert trajectories. These datasets are designed to test whether algorithms can leverage near-optimal data when it is partially present.

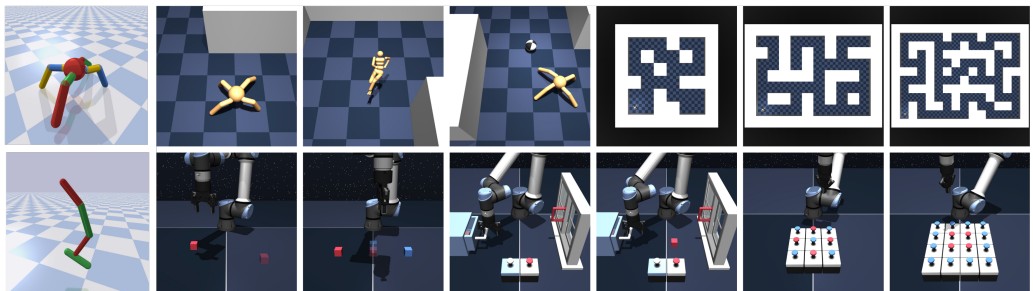

Figure 6: Visualization of offline RL tasks.

The AntMaze environments involve a quadruped ant navigating through a 2D maze using sparse goal-based rewards. The agent has a 29-dimensional state space and an 8-dimensional action space, corresponding to joint positions, velocities, and target location encoding. The tasks are particularly challenging due to long-horizon planning and sparse supervision.

- `antmaze-u` (umaze): A small maze where the agent must reach a fixed goal location using sparse rewards. The environment is relatively easy due to short trajectories.
- `antmaze-u-d` (umaze-diverse): Similar to `umaze`, but with broader trajectory diversity collected from random exploration.
- `antmaze-m-p` (medium-play): A medium-sized maze where data is collected via a play policy. The task is harder due to longer horizons and sparse goal rewards.
- `antmaze-m-d` (medium-diverse): Features more diverse and noisy behavior than `medium-play`, increasing exploration coverage but decreasing consistency.
- `antmaze-l-p` (large-play): A large maze with random play data. The agent must navigate long distances, making the task especially difficult under sparse reward signals.
- `antmaze-l-d` (large-diverse): Similar to `large-play`, but with broader and more varied behavior. It is one of the most challenging offline datasets due to the size of the environment and variability in data.

OGBench not only substantially extends the original AntMaze environment provided by the D4RL benchmark, but also introduces more challenging tasks, such as humanoid control and object manipulation. We elaborate on our selected environments below.

- `antmaze-giant-navigate`: An 8-DoF ant agent needs to reach a goal location in a 2-D maze, the size of which is substantially larger than those of D4RL datasets.
- `humanoid-medium/large-navigate`: This task involves full-body control of a 21-DoF Humanoid agent, which requires long-horizon reasoning.
- `antsoccer-arena-navigate`: This task involves controlling an ant agent to dribble a soccer ball. The agent is required to approach the ball and dribble it to random locations in an arena.
- `cube-single/double-play`: This task involves pick-and-place manipulation of several different cubes. The agent is required to complete tasks by moving, stacking, swapping, or permuting cubes. The options "single" and "double" refer to the number of cubes.
- `scene-play`: The agent's goal to manipulate the two buttons to determine the statuses of a window and a drawer to finish a specific task, which requires sequential reasoning.
- `puzzle-3×3/4×4-play`: The goal is to solve a "light-out" puzzle with a robot arm. At each step, the agent can press a button to change the states of the pressed button and its neighbors. We select $3 \times 3$ and $4 \times 4$ as the puzzle size.

**Network Architecture and hyperparameters.** In this part, we provide details of the hyperparameters used in our experiments on D4RL datasets and OGBench. Since most of the hyperparameters remain the same for all of our experiments, we list them in Table 4 and demonstrate our choice of

Table 4: Default Hyperparameters for VGF.

| Hyperparameter | Value |
|---|---|
| Critic learning rate | 0.0003 |
| Actor learning rate | 0.0003 |
| Gradient steps | 1000000 |
| Batch size | 256 |
| Critic Hidden dimensions | $[512, 512, 512, 512]$ |
| Discount factor $\gamma$ | 0.99 (default), 0.995 (antmaze-giant, humanoidmaze, antsoccer) |
| BC flow steps | 10 |
| Q value Aggregation | min (D4RL and OGBench antmaze), mean (OGBench others) |
| Particle number $N$ | 5 |
| Train particle select | mean |
| Eval particle select | max |
| Test VGF steps $L_{\text{test}}$ | $[0, 1, 2, 3]$ |

the important ones in Table 5 and Table 6. Specifically, we regard VGF step size $\epsilon$ and training VGF steps $L_{\text{train}}$ as the important hyperparameters.

We use a 4-layer MLP with 512 hidden units and Adam optimizer (Kingma & Ba, 2015) with a learning rate of $3 \times 10^{-4}$ for both the behavior cloning network and the critic network in all tasks. We also use a target network with soft update weight $5 \times 10^{-3}$ for critic update. We ran VGF for $10^6$ gradient steps for all our experiments with batch size 256. We set BC flow steps to 10. In addition, we set particle number $N$ to 5 because increasing $N$ will bring about extra computational costs but hardly any improvement in performance. Our results are averaged across 5 seeds for all our experiments, with standard deviation reported. In Table 7, we further provide detailed results on each of the $5 \times 9 = 45$ tasks in OGBench, which further verifies VGF's efficacy across most of the environments.

## B.2 RLHF EXPERIMENT DETAILS

We evaluate on the TL;DR Summarize corpus (Stiennon et al., 2020) and the Anthropic Helpful & Harmless (HH) corpus (Bai et al., 2022). To reduce degenerate generations that miss an eos token, we filter out overly long prompts before training and evaluation: we discard prompts longer than 448 tokens for TL;DR and 348 tokens for HH (lengths measured after tokenization with the base model's tokenizer). For TL;DR we use the dedicated SFT split provided by the dataset. Unless otherwise noted, both the reward model and the policy are initialized from the same SFT checkpoint. For Pythia models we train the SFT stage for 2 epochs with an initial learning rate of $\times 10^{-5}$ on both summarization and dialogue tasks. We train the reward model for 1 epoch with an initial learning rate of $1 \times 10^{-5}$ using the same train splits as the policy initialization (TL;DR references or HH chosen responses). For both SFT and reward model training we use cosine learning-rate decay with a warm-up ratio of 0.03. All other implementation details are kept identical across tasks unless explicitly stated.

Table 5: Hyperparameter selection of VGF in D4RL datasets.

| Env | Train VGF steps $L_{\text{train}}$ | VGF step size $\epsilon$ |
| --- | --- | --- |
| halfcheetah-medium-v2 | 3 | 0.05 |
| hopper-medium-v2 | 1 | 0.05 |
| walker2d-medium-v2 | 1 | 0.05 |
| halfcheetah-medium-replay-v2 | 3 | 0.05 |
| hopper-medium-replay-v2 | 1 | 0.05 |
| walker2d-medium-replay-v2 | 1 | 0.05 |
| halfcheetah-medium-expert-v2 | 3 | 0.05 |
| hopper-medium-expert-v2 | 1 | 0.05 |
| walker2d-medium-expert-v2 | 1 | 0.05 |
| antmaze-umaze-v2 | 5 | 0.2 |
| antmaze-umaze-diverse-v2 | 5 | 0.2 |
| antmaze-medium-play-v2 | 5 | 0.2 |
| antmaze-medium-diverse-v2 | 5 | 0.1 |
| antmaze-large-play-v2 | 5 | 0.2 |
| antmaze-large-diverse-v2 | 5 | 0.2 |

Table 6: Hyperparameter selection of VGF in OGBench datasets (offline and offline-to-online).

| Env | Train VGF steps $L_{\text{train}}$ | VGF step size $\epsilon$ |
| --- | --- | --- |
| antmaze-large-navigate-singletask-v0 | 5 | 0.1 |
| antmaze-giant-navigate-singletask-v0 | 5 | 0.1 |
| humanoidmaze-medium-navigate-singletask-v0 | 1 | 0.05 |
| humanoidmaze-large-navigate-singletask-v0 | 1 | 0.05 |
| antsoccer-arena-navigate-singletask-v0 | 2 | 0.05 |
| cube-single-play-singletask-v0 | 1 | 0.05 |
| cube-double-play-singletask-v0 | 1 | 0.05 |
| scene-play-singletask-v0 | 1 | 0.1 |
| puzzle-3x3-play-singletask-v0 | 5 | 0.1 |
| puzzle-4x4-play-singletask-v0 | 5 | 0.1 |

Table 7: OGBench results (all tasks). VGF performs comparable or superior to the baselines on most tasks. (*) denotes the default task per environment (Park et al., 2025b).

| Environment (5 tasks each) | Gaussian Policy | | | Diffusion/Flow Policy | | | w/o Policy (Ours) |
| --- | --- | --- | --- | --- | --- | --- | --- |
| | BC | IQL | ReBRAC | FBRAC | IDQL | FQL | VGF |
| antmaze-giant-task1(*) | $0_{\pm0}$ | $0_{\pm0}$ | $27_{\pm22}$ | $0_{\pm1}$ | $0_{\pm0}$ | $4_{\pm5}$ | $0_{\pm0}$ |
| antmaze-giant-task2 | $0_{\pm0}$ | $1_{\pm1}$ | $16_{\pm17}$ | $4_{\pm7}$ | $0_{\pm0}$ | $9_{\pm7}$ | $9_{\pm3}$ |
| antmaze-giant-task3 | $0_{\pm0}$ | $0_{\pm0}$ | $34_{\pm22}$ | $0_{\pm0}$ | $0_{\pm0}$ | $0_{\pm1}$ | $0_{\pm0}$ |
| antmaze-giant-task4 | $0_{\pm0}$ | $0_{\pm0}$ | $5_{\pm12}$ | $9_{\pm4}$ | $0_{\pm0}$ | $14_{\pm23}$ | $0_{\pm0}$ |
| antmaze-giant-task5 | $1_{\pm1}$ | $19_{\pm7}$ | $49_{\pm22}$ | $6_{\pm10}$ | $0_{\pm1}$ | $16_{\pm28}$ | $6_{\pm2}$ |
| hmmaze-medium-task1(*) | $1_{\pm0}$ | $32_{\pm7}$ | $16_{\pm9}$ | $25_{\pm8}$ | $1_{\pm1}$ | $19_{\pm12}$ | $86_{\pm1}$ |
| hmmaze-medium-task2 | $1_{\pm0}$ | $41_{\pm9}$ | $18_{\pm16}$ | $76_{\pm10}$ | $1_{\pm1}$ | $94_{\pm3}$ | $92_{\pm2}$ |
| hmmaze-medium-task3 | $6_{\pm2}$ | $25_{\pm5}$ | $36_{\pm13}$ | $27_{\pm11}$ | $0_{\pm1}$ | $74_{\pm18}$ | $87_{\pm2}$ |
| hmmaze-medium-task4 | $0_{\pm0}$ | $0_{\pm1}$ | $15_{\pm16}$ | $1_{\pm2}$ | $1_{\pm1}$ | $3_{\pm4}$ | $0_{\pm0}$ |
| hmmaze-medium-task5 | $2_{\pm1}$ | $66_{\pm4}$ | $24_{\pm20}$ | $63_{\pm9}$ | $1_{\pm1}$ | $97_{\pm2}$ | $97_{\pm0}$ |
| hmmaze-large-task1(*) | $0_{\pm0}$ | $3_{\pm1}$ | $2_{\pm1}$ | $0_{\pm1}$ | $0_{\pm0}$ | $7_{\pm6}$ | $26_{\pm4}$ |
| hmmaze-large-task2 | $0_{\pm0}$ | $0_{\pm0}$ | $0_{\pm0}$ | $0_{\pm0}$ | $0_{\pm0}$ | $0_{\pm0}$ | $2_{\pm1}$ |
| hmmaze-large-task3 | $1_{\pm1}$ | $7_{\pm3}$ | $8_{\pm4}$ | $10_{\pm2}$ | $3_{\pm1}$ | $11_{\pm7}$ | $12_{\pm6}$ |
| hmmaze-large-task4 | $1_{\pm0}$ | $1_{\pm0}$ | $1_{\pm1}$ | $0_{\pm0}$ | $0_{\pm0}$ | $2_{\pm3}$ | $2_{\pm1}$ |
| hmmaze-large-task5 | $0_{\pm1}$ | $1_{\pm1}$ | $2_{\pm2}$ | $1_{\pm1}$ | $0_{\pm0}$ | $1_{\pm3}$ | $31_{\pm5}$ |
| antsoccer-arena-task1 | $2_{\pm1}$ | $14_{\pm5}$ | $0_{\pm0}$ | $17_{\pm3}$ | $44_{\pm12}$ | $77_{\pm4}$ | $76_{\pm3}$ |
| antsoccer-arena-task2 | $2_{\pm2}$ | $17_{\pm7}$ | $0_{\pm1}$ | $8_{\pm2}$ | $15_{\pm12}$ | $88_{\pm3}$ | $75_{\pm2}$ |
| antsoccer-arena-task3 | $0_{\pm0}$ | $6_{\pm4}$ | $0_{\pm0}$ | $16_{\pm3}$ | $0_{\pm0}$ | $61_{\pm6}$ | $41_{\pm9}$ |
| antsoccer-arena-task4(*) | $1_{\pm0}$ | $3_{\pm2}$ | $0_{\pm0}$ | $24_{\pm4}$ | $0_{\pm1}$ | $39_{\pm6}$ | $58_{\pm8}$ |
| antsoccer-arena-task5 | $0_{\pm0}$ | $2_{\pm2}$ | $0_{\pm0}$ | $15_{\pm4}$ | $0_{\pm0}$ | $36_{\pm9}$ | $64_{\pm8}$ |
| cube-single-task1 | $10_{\pm5}$ | $88_{\pm3}$ | $89_{\pm5}$ | $73_{\pm33}$ | $95_{\pm2}$ | $97_{\pm2}$ | $98_{\pm2}$ |
| cube-single-task2(*) | $3_{\pm1}$ | $85_{\pm8}$ | $92_{\pm4}$ | $83_{\pm13}$ | $96_{\pm2}$ | $97_{\pm2}$ | $100_{\pm0}$ |
| cube-single-task3 | $9_{\pm3}$ | $91_{\pm5}$ | $93_{\pm3}$ | $82_{\pm12}$ | $99_{\pm1}$ | $98_{\pm2}$ | $100_{\pm0}$ |
| cube-single-task4 | $2_{\pm1}$ | $73_{\pm6}$ | $92_{\pm3}$ | $79_{\pm20}$ | $93_{\pm4}$ | $94_{\pm3}$ | $94_{\pm3}$ |
| cube-single-task5 | $3_{\pm3}$ | $78_{\pm9}$ | $87_{\pm8}$ | $76_{\pm33}$ | $90_{\pm6}$ | $93_{\pm3}$ | $92_{\pm4}$ |
| cube-double-task1 | $8_{\pm3}$ | $27_{\pm5}$ | $45_{\pm6}$ | $47_{\pm11}$ | $39_{\pm19}$ | $61_{\pm9}$ | $95_{\pm6}$ |
| cube-double-task2(*) | $0_{\pm0}$ | $1_{\pm1}$ | $7_{\pm3}$ | $22_{\pm12}$ | $16_{\pm10}$ | $36_{\pm6}$ | $78_{\pm10}$ |
| cube-double-task3 | $0_{\pm0}$ | $0_{\pm0}$ | $4_{\pm1}$ | $4_{\pm2}$ | $17_{\pm8}$ | $22_{\pm5}$ | $66_{\pm8}$ |
| cube-double-task4 | $0_{\pm0}$ | $0_{\pm0}$ | $1_{\pm1}$ | $0_{\pm1}$ | $0_{\pm1}$ | $5_{\pm2}$ | $31_{\pm4}$ |
| cube-double-task5 | $0_{\pm0}$ | $4_{\pm3}$ | $4_{\pm2}$ | $2_{\pm2}$ | $1_{\pm1}$ | $19_{\pm10}$ | $78_{\pm11}$ |
| scene-task1 | $19_{\pm6}$ | $94_{\pm3}$ | $95_{\pm2}$ | $96_{\pm8}$ | $100_{\pm0}$ | $100_{\pm0}$ | $100_{\pm0}$ |
| scene-task2(*) | $1_{\pm1}$ | $12_{\pm3}$ | $50_{\pm13}$ | $46_{\pm10}$ | $33_{\pm14}$ | $76_{\pm9}$ | $96_{\pm2}$ |
| scene-task3 | $1_{\pm1}$ | $32_{\pm7}$ | $55_{\pm16}$ | $78_{\pm14}$ | $94_{\pm4}$ | $98_{\pm1}$ | $98_{\pm2}$ |
| scene-task4 | $2_{\pm2}$ | $0_{\pm1}$ | $3_{\pm3}$ | $4_{\pm4}$ | $4_{\pm3}$ | $5_{\pm1}$ | $2_{\pm2}$ |
| scene-task5 | $0_{\pm0}$ | $0_{\pm0}$ | $0_{\pm0}$ | $0_{\pm0}$ | $0_{\pm0}$ | $0_{\pm0}$ | $2_{\pm1}$ |
| puzzle-3x3-task1 | $5_{\pm2}$ | $33_{\pm6}$ | $97_{\pm4}$ | $63_{\pm19}$ | $52_{\pm12}$ | $90_{\pm4}$ | $100_{\pm0}$ |
| puzzle-3x3-task2 | $1_{\pm1}$ | $4_{\pm3}$ | $1_{\pm1}$ | $2_{\pm2}$ | $0_{\pm1}$ | $16_{\pm5}$ | $71_{\pm6}$ |
| puzzle-3x3-task3 | $1_{\pm1}$ | $3_{\pm2}$ | $3_{\pm1}$ | $1_{\pm1}$ | $0_{\pm0}$ | $10_{\pm3}$ | $62_{\pm7}$ |
| puzzle-3x3-task4(*) | $1_{\pm1}$ | $2_{\pm1}$ | $2_{\pm1}$ | $2_{\pm2}$ | $0_{\pm0}$ | $16_{\pm5}$ | $65_{\pm6}$ |
| puzzle-3x3-task5 | $1_{\pm0}$ | $3_{\pm2}$ | $5_{\pm3}$ | $2_{\pm2}$ | $0_{\pm0}$ | $16_{\pm3}$ | $76_{\pm6}$ |
| puzzle-4x4-task1 | $1_{\pm1}$ | $12_{\pm2}$ | $26_{\pm4}$ | $32_{\pm9}$ | $48_{\pm5}$ | $34_{\pm8}$ | $85_{\pm8}$ |
| puzzle-4x4-task2 | $0_{\pm0}$ | $7_{\pm4}$ | $12_{\pm4}$ | $5_{\pm3}$ | $14_{\pm5}$ | $16_{\pm5}$ | $26_{\pm3}$ |
| puzzle-4x4-task3 | $0_{\pm0}$ | $9_{\pm3}$ | $15_{\pm3}$ | $20_{\pm10}$ | $34_{\pm5}$ | $18_{\pm5}$ | $65_{\pm6}$ |
| puzzle-4x4-task4(*) | $0_{\pm0}$ | $5_{\pm2}$ | $10_{\pm3}$ | $5_{\pm1}$ | $26_{\pm6}$ | $11_{\pm3}$ | $33_{\pm6}$ |
| puzzle-4x4-task5 | $0_{\pm0}$ | $4_{\pm1}$ | $7_{\pm3}$ | $4_{\pm3}$ | $24_{\pm11}$ | $7_{\pm3}$ | $18_{\pm2}$ |

## C  PSEDOCODE OF VGF

```python
import jax
import jax.numpy as jnp

def rbf_kernel(X, Y, sigma=None):
    """X: [B, n, d], Y: [B, m, d], returns K_XY: [B, n, m]"""
    X2 = jnp.sum(X * X, axis=-1, keepdims=True)
    Y2 = jnp.sum(Y * Y, axis=-1, keepdims=True).transpose(0, 2, 1)
    XY = jnp.matmul(X, Y.transpose(0, 2, 1))
    dnorm2 = X2 + Y2 - 2.0 * XY
    dnorm2 = jnp.maximum(dnorm2, 0.0)
    if sigma is None:
        # median heuristic per batch
        h = jnp.median(dnorm2, axis=(1,2))
        h /= (2.0 * jnp.log(X.shape[1] + 1.0))
        sigma_val = jnp.sqrt(jnp.maximum(h, 1e-12))
        sigma_val = sigma_val[:, None, None]
    else:
        sigma_val = jnp.asarray(sigma)
        if sigma_val.ndim == 0:
            sigma_val = jnp.broadcast_to(sigma_val, (X.shape[0], 1, 1))
    gamma = 1.0 / (1e-6 + 2.0 * (sigma_val ** 2))
    K_XY = jnp.exp(-gamma * dnorm2)
    return K_XY

class VGF:
    def __init__(self, q, alpha, optimizer):
        self.q = q
        self.alpha = alpha
        self.optim = optimizer
        self.opt_state = None

    def init(self, particles):
        self.opt_state = self.optim.init(particles)
        return particles, self.opt_state

    def phi(self, obs, particles):
        # obs: [B, D], particles: [B, N, D]
        # score terms
        def sum_q(action):
            obs_flatten = obs.reshape(-1, obs.shape[-1])
            action_flatten = action.reshape(-1, action.shape[-1])
            qs = self.q(obs_flatten, action_flatten)
            q = jnp.mean(qs, axis=0)
            return jnp.sum(q)
        score = jax.grad(sum_q)(particles)
        # kernel terms
        particles_stop = jax.lax.stop_gradient(particles)
        K_xx = rbf_kernel(particles, particles_stop)
        def sum_K(x):
            return jnp.sum(rbf_kernel(x, particles_stop))
        grad_q = jax.lax.stop_gradient(K_xx) @ score
        grad_K = -jax.grad(sum_K)(particles)
        phi_val = (grad_q / self.alpha + grad_K) / particles.shape[1]
        return phi_val

    def step(self, obs, particles, opt_s):
        grads = self.phi(obs, particles)
        updates, new_opt_s = self.optim.update(-grads, opt_s, particles)
        new_particles = optax.apply_updates(particles, updates)
        return new_particles, new_opt_s
```

Figure 7: A simple implementation of the VGF process.

