# OpenReview forum: "Reinforcement Learning via Value Gradient Flow"
_ICLR.cc/2026/Conference — ICLR 2026 Poster_

### Official Review · Reviewer_Y9ry · 2025-10-20

**Soundness:** 3
**Presentation:** 2
**Contribution:** 3
**Rating:** 6
**Confidence:** 3

**Summary:**

In this paper, an offline reinforcement learning framework is formulated as an optimal transport problem that transfers the actions induced by the behavior policy to those induced by the optimal policy.
This transport process is modeled as a particle-based gradient flow, and the resulting action update rule is analogous to Stein Variational Gradient Descent (SVGD).

In existing methods, a regularization term is typically introduced into the objective function, and its coefficient must be carefully tuned to balance the trade-off between maximizing the expected return and enforcing behavior regularization.
In contrast, the proposed method controls the degree of regularization through the transport budget. Specifically, the number of steps and the learning rate.

The proposed approach is evaluated on continuous control tasks in D4RL and OGBench, as well as on LLM fine-tuning tasks using TL;DR and Anthropic-HH datasets.
The experimental results demonstrate that the proposed method outperforms existing approaches on these benchmark tasks.
In addition, a toy example illustrates that the method can successfully model a multimodal action distribution.

**Strengths:**

- Formulating offline RL as an optimal transport problem is both interesting and appears to be novel.

- The experimental results clearly show the advantages of the proposed method in both continuous control and LLM fine-tuning tasks.

Although the idea of using the Wasserstein distance as a form of regularization has been explored in prior offline RL studies, I am not aware of previous work that learns a gradient flow from the behavior policy to the optimal policy.
In this regard, the proposed method is novel and conceptually appealing.

**Weaknesses:**

- The presentation is somewhat unclear in several places and would benefit from improvement.

- The paper lacks any discussion of computational cost.

The claim that “VGF removes the need to balance the optimization conflict between reward maximization and deviation penalties” is somewhat misleading. While it is true that the proposed method eliminates the need to manually tune the coefficient of an explicit regularization term, it still requires tuning the transport budget, which effectively serves a similar role. Thus, balancing between reward maximization and deviation control remains necessary. Since similar statements appear multiple times throughout the paper, the authors should carefully revise these passages to avoid confusion.

The action update rule in Equation (7) is analogous to SVGD. However, although Liu & Wang (2016) and Liu (2017) are cited, the term SVGD itself does not appear explicitly. Explicitly mentioning SVGD would make the connection clearer and help readers understand the proposed algorithm more easily.

Another potential weakness of the proposed method lies in its computational cost. Action generation using the flow-based behavior policy introduces additional overhead, and the gradient flow from the behavior policy to the optimal policy also incurs computational expense.
I expect that the proposed method would be significantly slower than TD3+BC and IQL, and possibly even slower than Diffusion-QL.
However, the JAX-based implementation may help alleviate some of these costs. Discussing these computational limitations would strengthen the overall presentation and contextualize the method’s contributions.

**Questions:**

- Please comment on the statement regarding the removal of the need to balance between reward maximization and deviation penalties. If you agree with my observation, please revise the paper accordingly.

- Why is SVGD not explicitly mentioned in the paper? Please elaborate on the connection between the proposed method and SVGD.

- How much training time does the proposed method require in practice? Please provide approximate wall-clock times and compare them against the baseline methods.

**Details Of Ethics Concerns:**

I am also reviewing Submission 10969, titled “Iterative Refinement of Flow Policies in Probability Space for Online Reinforcement Learning.”

I noticed that the idea presented in this paper is highly similar to that of Submission 691. Both submissions share the same main approach: a flow policy is initialized using a given dataset to model the behavior policy, and the transport from the actions generated by the initial policy to those of the optimal policy is learned. The optimal transport is modeled using the Jordan Kinderlehrer Otto (JKO) scheme.

There are, however, some differences between the two submissions.
In Submission 691, the update rule, which is analogous to Stein Variational Gradient Descent, is clearly described, and the regularization based on limiting the transport budget is explained. In addition, the gradient flow appears to be modeled using a single model. The method is evaluated on both locomotion and large language model (LLM) fine tuning tasks.
In contrast, in Submission 10969, the update rule is not clearly described, and the transport is modeled with step wise models. The method is evaluated on manipulation tasks.

In my view, the differences between these two papers are minimal. If they are authored by the same group, these submissions may constitute a dual submission of essentially the same work. However, if the authors are different, then this overlap is less concerning. As I do not have access to the author identities, I am simply raising a flag to alert the Area Chairs and Program Chairs to this potential issue. I have also raised a similar flag for Submission 10969.

---

> ### Author Response · Authors · 2025-11-22
>
> We thank the reviewer for the effort engaged in the review phase and the constructive comments. Regarding the concerns, we provide the detailed responses separately as follows.
>
>
> > The claim that “VGF removes the need to balance the optimization conflict between reward maximization and deviation penalties” is somewhat misleading. While it is true that the proposed method eliminates the need to manually tune the coefficient of an explicit regularization term, it still requires tuning the transport budget, which effectively serves a similar role. Thus, balancing between reward maximization and deviation control remains necessary. Since similar statements appear multiple times throughout the paper, the authors should carefully revise these passages to avoid confusion.
>
> Thanks for pointing that out. We agree that VGF still uses a transport budget, which acts as an implicit regularization term. We will revise the sentences to avoid confusion:
>
> `“VGF suffers less from balancing the optimization conflict between reward maximization and deviation penalties.”`
>
>
>
> > Please elaborate on the connection between the proposed method and SVGD.
>
> Thanks for pointing that out. We do cite SVGD in the paper, and we do not claim it is a novel method introduced by us. The contribution of our paper is framing behavior-regularized RL as an optimal transport problem, where SVGD serves merely as a practical solution, and many other solvers could also be applied. Note that a key difference between VGF and SVGD is that VGF uses particles from a reference distribution instead of Gaussian noise, requiring fewer particles due to the inductive bias provided by the reference distribution.
>
>
>
>
> > How much training time does the proposed method require in practice? Please provide approximate wall-clock times and compare them against the baseline methods.
>
> Thanks for pointing this out. We provide the training cost, inference cost, and memory usage of representative baselines across various environments. Specifically, we compare VGF with IQL (Gaussian policy), DQL (diffusion policy), IFQL (flow policy), and FQL (one-step flow policy).
>
> The training and inference costs are measured by the time per step under the default set of parameters. Note that VGF uses smaller flow/diffusion steps (5) than Diffusion-QL, so it reduces the inference cost, while we found that Diffusion-QL with fewer steps performs much worse than the original setting.
> For memory usage, we disable JAX memory preallocation to allow a fair comparison of GPU memory usage.
>
> We find that VGF has the lowest training cost and the least memory usage. For inference cost, VGF is consistently faster than DQL and IFQL, while slightly slower than FQL and IQL. We believe it is reasonable to sacrifice a small amount of inference compute to obtain a significant performance boost.
>
>
> | Environment | Metric         | VGF    | Diffusion-QL | FQL    | IFQL   | IQL    |
> |------------|----------------|--------|---------------------|--------|--------|--------|
> | halfcheetah-m-e | Training time  | 1.92ms | 2.11ms | 2.10ms | 2.16ms | 2.04ms |
> | halfcheetah-m-e | Inference time | 0.63ms | 1.18ms | 0.32ms | 0.94ms | 0.36ms |
> | halfcheetah-m-e | Memory usage   | 484MB  | 496MB  | 498MB  | 498MB  | 498MB  |
> | antmaze-m-d    | Training time  | 1.94ms | 2.36ms | 2.16ms | 2.22ms | 2.15ms |
> | antmaze-m-d    | Inference time | 0.71ms | 1.44ms | 0.37ms | 0.93ms | 0.34ms |
> | antmaze-m-d    | Memory usage   | 488MB  | 500MB  | 500MB  | 500MB  | 498MB  |
> | humanoidmaze-m    | Training time  | 2.71ms | 3.28ms | 2.78ms | 2.92ms | 3.02ms |
> | humanoidmaze-m   | Inference time | 0.81ms | 1.54ms | 0.59ms | 0.98ms | 0.47ms |
> | humanoidmaze-m    | Memory usage   | 640MB  | 648MB  | 654MB  | 654MB  | 652MB  |
> | hopper-m     | Training time  | 2.18ms | 2.54ms | 2.14ms | 2.21ms | 2.25ms |
> | hopper-m     | Inference time | 0.67ms | 1.24ms | 0.38ms | 0.88ms | 0.41ms |
> | hopper-m      | Memory usage   | 484MB  | 494MB  | 496MB  | 496MB  | 492MB  |
> | walker2d-m     | Training time  | 2.07ms | 2.47ms | 2.20ms | 2.26ms | 2.44ms |
> | walker2d-m     | Inference time | 0.65ms | 1.24ms | 0.39ms | 0.93ms | 0.39ms |
> | walker2d-m     | Memory usage   | 484MB  | 496MB  | 500MB  | 498MB  | 498MB  |
>
> > About the dual submission.
>
> Thank you for bringing this to our attention. We would like to clarify that we are not aware of Submission 10969, any similarity in high-level intuition is entirely coincidental.
> In particular, our method uses a minimalistic value-guided flow update without the step-wise modeling or architectural complexities described in the other submission. The overall algorithmic structure, training objective, and implementation pipeline differ substantially, with our approach to be much simpler, more elegant, and more lightweight, while still achieving strong empirical performance.
>
>
>
> Please let us know if any further questions remain. We hope the reviewer can reassess our work with these clarifications.

---

> > ### Comment · Area_Chair_Vxrv · 2025-11-26
> > **AC Triage Decision for Ethics Flag on Submission 691**
> >
> > Dear Ethics Chairs,
> >
> > Thank you to the reviewer for raising the question and ensuring careful oversight. After thoroughly reviewing both Submission 691 and Submission 10969, I have determined that the concern does not constitute a dual submission or ethics violation. Although both papers employ the general JKO / Wasserstein gradient-flow formalism—a standard and broadly used framework—their technical content, problem settings, algorithms, and empirical contributions are clearly distinct.
> >
> > Submission 691 introduces an implicit, particle-based value-gradient flow method for behavior-regularized offline RL and RLHF, where the flow model functions only as a sampler and the policy itself is not parameterized as a flow. Submission 10969, in contrast, presents a parametric stepwise flow-policy optimization framework for online fine-tuning of pretrained flow-matching policies in robotic control, using block-wise JKO updates and explicit Wasserstein trust-region regularization. The algorithmic mechanisms, architectures, objectives, update rules, and evaluations differ substantially, and there is no reuse of text, figures, or experiments.
> >
> > Based on this assessment, the similarity reflects conceptual relatedness within a shared theoretical lens, not dual submission. This matter can be fully addressed at the AC level and does not require a separate ethics review.
> >
> > Best regards,
> >
> > Area Chair, ICLR 2026

---

> > > ### Comment · Program_Chairs · 2025-11-27
> > > **Confirming No Overlap in the Authorship**
> > >
> > > Confirming that the two submissions do not share authors.

---

> ### Comment · Reviewer_Y9ry · 2025-11-27
>
> Thank you for your detailed response. Regarding the balance between reward maximization and deviation penalties, could you clarify why VGF suffers less compared to other algorithms that include an explicit regularization term?
> From my perspective, tuning the three hyperparameters, train time flow steps $L_\text{train}$, temperature $\alpha$, and particle number $N$, does not seem easier than tuning a coefficient for an explicit regularization term in other algorithms.
>
> I appreciate the additional result on computation costs and the comparison with baseline methods. It seems reasonable that VGF requires more inference time than FQL, given that FQL employs a one step policy.
>
> Regarding the relation to submission 10969, thank you for the clarification. Since the authors are not overlapping, I have no further concerns on this point.

---

### Official Review · Reviewer_gCX9 · 2025-10-30

**Soundness:** 4
**Presentation:** 3
**Contribution:** 3
**Rating:** 8
**Confidence:** 4

**Summary:**

This manuscript proposes the Value Gradient Flow (VGF) for  behavior-regularized reinforcement learning. VGF formulates behavior-regularized RL as an optimal transport problem, guiding particles from the reference distribution (offline data or supervised fine-tuned policy) toward high-value regions via discrete gradient flow. The VGF framework demonstrates several merits including avoiding unnecessary restrictions on optimization problem, no need explicit policy parameterization and so on. Experiments on D4RL and other benchmarks demonstrate the superiority of the proposed method.

**Strengths:**

* It is appreciated that the authors combine the idea of optimal transport, a classic method in optimization, with RL situations.

* The first strength of this manuscript is that it avoid optimization conflict between reward maximization and distribution penalty, mitigates the "deadly triad" problem in actor-critic algorithms, and enables more stable training.

* The idea of applying the latent variable in generative modeling is appealing. It combines the good merits in generative modeling (dimension reduction and re-parametrization), and applies well in exploration and exploitation.

**Weaknesses:**

1. We wonder whether the performance of VGF is limited by the reference distribution. When the reference distribution is skewed towards to suboptimal behaviors, how is the robustness of VGF?

2. Particle-based gradient flow solving requires maintaining multiple particles and function calculations, do the authors consider such cost in practice? Especially in high dimension scenarios

3. From the experiments, it seems the VGF still inferior to Guassian Policy or Diffusion policy in some D4RL, it is suggsted the authors provide additional explanations on this issue.

4. Although penalty coefficient tuning is unnecessary, key hyper-parameters such as training steps and temperature still require manual task-specific adjustment, lacking adaptive selection mechanisms. We encourage the authors do more trial in this field.

**Questions:**

See the weakness above

---

> ### Author Response · Authors · 2025-11-22
>
> We thank the reviewer for the effort engaged in the review phase and the constructive comments.
>
>
> > We wonder whether the performance of VGF is limited by the reference distribution. When the reference distribution is skewed towards to suboptimal behaviors, how is the robustness of VGF?
>
> Thanks for pointing that out — that’s a great question. Note that several datasets in offline RL benchmarks already contain mixed or suboptimal behaviors, such as the medium and medium-replay datasets in D4RL, and VGF achieves strong results on them. For datasets that are highly skewed toward suboptimal behavior, we believe this remains an open challenge for behavior-regularized RL more broadly.
>
> > Particle-based gradient flow solving requires maintaining multiple particles and function calculations, do the authors consider such cost in practice? Especially in high dimension scenarios
>
> Thanks for pointing that out. Please see our responses for Reviewer 1 or 2, where we provide training cost, inference cost, and memory usage of representative baselines across various environments, including high-dimensional tasks such as Antmaze (dim = 8) and Humanoidmaze (dim = 21).
>
>
> > From the experiments, it seems the VGF still inferior to Guassian Policy or Diffusion policy in some D4RL, it is suggsted the authors provide additional explanations on this issue.
>
> Thanks for pointing that out. VGF is only inferior to Gaussian policies or diffusion policies in some medium-expert datasets. These datasets are relatively easy because expert data is already included. Our suspicion is that some hyperparameters of the Flow BC policy may need tuning (e.g., learning rate, flow steps) for these datasets, as FQL also does not perform well. In our results, we follow the default settings in the FQL codebase, which were originally designed for OGBench datasets.
>
>
>
> > Although penalty coefficient tuning is unnecessary, key hyper-parameters such as training steps and temperature still require manual task-specific adjustment, lacking adaptive selection mechanisms. We encourage the authors do more trial in this field.
>
> Thanks for pointing that out. It is true that several parameters require tuning for new environments. However, we argue that only the VGF step L and learning rate \epsilon require careful tuning, while other parameters, including the temperature \alpha, have much less impact. Our robustness results for \epsilon and \alpha support this claim. Therefore, we recommend tuning L and \epsilon first before adjusting other parameters.
>
> | Robustness of \epsilon         | 0.01  | 0.05  | 0.10  | 0.20  | 0.50  |
> |-----------------|-------|-------|-------|-------|-------|
> | halfcheetah-m-e | 89.55 | 87.44 | 89.49 | 75.94 | 44.32 |
> | hopper-m-e      | 106.40| 98.12 | 58.79 | 26.98 | 16.50 |
> | antmaze-m-d     | 2     | 20    | 68    | 64    | 60    |
> | walker2d-m      | 80.26 | 85.21 | 84.98 | 80.14 | 57.41 |
> | hopper-m        | 90.07 | 91.66 | 83.71 | 73.87 | 48.46 |
>
> | Robustness of \alpha         | 0.01  | 0.10  | 0.50  | 1.00  | 2.00  |
> |-----------------|-------|-------|-------|-------|-------|
> | halfcheetah-m-e | 86.06 | 87.51 | 87.82 | 89.49 | 89.87 |
> | walker2d-m      | 85.82 | 86.52 | 85.98 | 84.98 | 84.80 |
> | antmaze-m-d     | 68    | 74    | 72    | 68    | 66    |
> | hopper-m-e      | 81.34 | 89.22 | 101.78| 98.45 | 94.25 |
> | hopper-m        | 88.54 | 90.81 | 94.74 | 91.72 | 91.17 |

---

> > ### Comment · Reviewer_gCX9 · 2025-11-24
> >
> > Thank you for the authors' nice reply. The rebuttal solves my concerns and I will keep my scores unchanged.

---

### Official Review · Reviewer_A97B · 2025-10-31

**Soundness:** 3
**Presentation:** 2
**Contribution:** 3
**Rating:** 4
**Confidence:** 3

**Summary:**

This paper proposes Value Gradient Flow, which solves the behavior regularized RL by initializing particles using the reference distribution and transporting the particles towards the target distribution, following the gradient of the value function. Compared to a vanilla value-guided MCMC, VGF also incorporates several other ingredients and techniques, such as the best-of-N refinement during evaluation and kernel-based affinity metrics. The evaluation of VGF is conducted on both offline RL and RLHF tasks. In Offline RL, VGF is used both for training and evaluation, while in RLHF, VGF is only used for test-time generation. The key findings are that 1) with proper hyperparameter configuration, VGF outperforms the flow-based actor critic method both on D4RL and OGBench; 2) On TL;DR and Anthropic-HH datasets, VGF seems to be efficient for alignment and outperforms baseline methods, including PPO, DPO by a large margin.

**Strengths:**

1. Instead of using a flow model to generate samples towards the target distribution, VGF instead first trains a flow model to initialize particles following a reference distribution and afterwards transports them towards high-valued areas. This approach is novel.

2. Besides, the VGF framework seems to unify several existing practices. For example, using zero transportation steps corresponds to best-of-N sampling. Conversely, as the number of transportation steps approaches infinity, the VGF process effectively recovers gradient-based sampling methods, such as Langevin dynamics.

**Weaknesses:**

1. VGF incurs a higher inference cost than traditional flow-based methods. This is because, in addition to the initial sample generation from the reference distribution, VGF requires a computationally intensive iterative process to transport and refine these particles toward high-density regions.

2. Furthermore, VGF appears highly sensitive to its hyperparameters and can demonstrate inconsistent performance trends across different tasks. Consequently, applying VGF to a new problem often requires extensive, task-specific tuning to achieve optimal results.

**Questions:**

1. In Table 3, is VGF using particles that the SFT model initializes?

2. Could the authors provide a runtime analysis of both VGF and other diffusion-based methods?

---

> ### Author Response · Authors · 2025-11-22
>
> We thank the reviewer for the effort engaged in the review phase and the constructive comments. Regarding the concerns, we provide the detailed responses separately as follows.
>
>
> > VGF incurs a higher inference cost than traditional flow-based methods. This is because, in addition to the initial sample generation from the reference distribution, VGF requires a computationally intensive iterative process to transport and refine these particles toward high-density regions.
>
>
>
> Thanks for pointing this out. We provide the training cost, inference cost, and memory usage of representative baselines across various environments. Specifically, we compare VGF with IQL (Gaussian policy), DQL (diffusion policy), IFQL (flow policy), and FQL (one-step flow policy).
>
> The training and inference costs are measured by the time per step under the default set of parameters. Note that VGF uses smaller flow/diffusion steps (5) than Diffusion-QL, so it reduces the inference cost, while we found that Diffusion-QL with fewer steps performs much worse than the original setting.
> For memory usage, we disable JAX memory preallocation to allow a fair comparison of GPU memory usage.
>
> We find that VGF has the lowest training cost and the least memory usage. For inference cost, VGF is consistently faster than DQL and IFQL, while slightly slower than FQL and IQL. We believe it is reasonable to sacrifice a small amount of inference compute to obtain a significant performance boost.
>
>
> | Environment | Metric         | VGF    | Diffusion-QL | FQL    | IFQL   | IQL    |
> |------------|----------------|--------|---------------------|--------|--------|--------|
> | halfcheetah-m-e | Training time  | 1.92ms | 2.11ms | 2.10ms | 2.16ms | 2.04ms |
> | halfcheetah-m-e | Inference time | 0.63ms | 1.18ms | 0.32ms | 0.94ms | 0.36ms |
> | halfcheetah-m-e | Memory usage   | 484MB  | 496MB  | 498MB  | 498MB  | 498MB  |
> | antmaze-m-d    | Training time  | 1.94ms | 2.36ms | 2.16ms | 2.22ms | 2.15ms |
> | antmaze-m-d    | Inference time | 0.71ms | 1.44ms | 0.37ms | 0.93ms | 0.34ms |
> | antmaze-m-d    | Memory usage   | 488MB  | 500MB  | 500MB  | 500MB  | 498MB  |
> | humanoidmaze-m    | Training time  | 2.71ms | 3.28ms | 2.78ms | 2.92ms | 3.02ms |
> | humanoidmaze-m   | Inference time | 0.81ms | 1.54ms | 0.59ms | 0.98ms | 0.47ms |
> | humanoidmaze-m    | Memory usage   | 640MB  | 648MB  | 654MB  | 654MB  | 652MB  |
> | hopper-m     | Training time  | 2.18ms | 2.54ms | 2.14ms | 2.21ms | 2.25ms |
> | hopper-m     | Inference time | 0.67ms | 1.24ms | 0.38ms | 0.88ms | 0.41ms |
> | hopper-m      | Memory usage   | 484MB  | 494MB  | 496MB  | 496MB  | 492MB  |
> | walker2d-m     | Training time  | 2.07ms | 2.47ms | 2.20ms | 2.26ms | 2.44ms |
> | walker2d-m     | Inference time | 0.65ms | 1.24ms | 0.39ms | 0.93ms | 0.39ms |
> | walker2d-m     | Memory usage   | 484MB  | 496MB  | 500MB  | 498MB  | 498MB  |
>
>
>
> > Furthermore, VGF appears highly sensitive to its hyperparameters, ..., applying VGF to a new problem often requires extensive, task-specific tuning to achieve optimal results.
>
>
> It is true that there are several parameters that should be tuned to achieve optimal results in a new environment. However, we argue that among all the parameters, only the VGF step L and the learning rate \epsilon require careful tuning, while the other parameters, including the temperature \alpha, have much less impact on the results. To verify this, we conducted additional experiments on the robustness of \epsilon and \alpha. The first table shows that the optimal learning rate differs across environments, implying the need to tune \epsilon. The second table indicates that, in most cases, different choices of \alpha do not lead to large performance differences. Therefore, the need to tune \alpha is less pressing than the need to tune L and \epsilon. We thus recommend tuning L and \epsilon first in a new environment before adjusting other hyperparameters.
>
> | Robustness of \epsilon         | 0.01  | 0.05  | 0.10  | 0.20  | 0.50  |
> |-----------------|-------|-------|-------|-------|-------|
> | halfcheetah-m-e | 89.55 | 87.44 | 89.49 | 75.94 | 44.32 |
> | hopper-m-e      | 106.40| 98.12 | 58.79 | 26.98 | 16.50 |
> | antmaze-m-d     | 2     | 20    | 68    | 64    | 60    |
> | walker2d-m      | 80.26 | 85.21 | 84.98 | 80.14 | 57.41 |
> | hopper-m        | 90.07 | 91.66 | 83.71 | 73.87 | 48.46 |
>
> | Robustness of \alpha         | 0.01  | 0.10  | 0.50  | 1.00  | 2.00  |
> |-----------------|-------|-------|-------|-------|-------|
> | halfcheetah-m-e | 86.06 | 87.51 | 87.82 | 89.49 | 89.87 |
> | walker2d-m      | 85.82 | 86.52 | 85.98 | 84.98 | 84.80 |
> | antmaze-m-d     | 68    | 74    | 72    | 68    | 66    |
> | hopper-m-e      | 81.34 | 89.22 | 101.78| 98.45 | 94.25 |
> | hopper-m        | 88.54 | 90.81 | 94.74 | 91.72 | 91.17 |
>
>
>
> > In Table 3, is VGF using particles that the SFT model initializes?
>
> Yes
>
>
> Please let us know if any further questions remain. We hope the reviewer can reassess our work with these clarifications.

---

> > ### Comment · Reviewer_A97B · 2025-11-22
> >
> > 1. I appreciate the response and additional results from the authors, which I believe should be included in the manuscript to provide the readers with a comprehensive understanding of VGF. However, I would like to note that there also exist methods [1] that employ diffusion policies while also achieving fast generation with a few denoising steps (e.g., 5 steps). Since VGF requires taking the gradient of value functions during inference, I would not expect it to have better inference speed than a plain diffusion policy. Besides, [1] seems to achieve better performance than VGF on D4RL locomotion tasks.
> >
> > 2. In the introduction (line 68), VGF is said to "balance the optimisation conflict between reward maximisation and derivation penalties, avoid brittle coefficient hyperparameter tuning and better preserve the multimodal structure". However, VGF actually balances the conflict by controlling the transportation budget, which also demonstrates sensitivity, as shown in the ablation study regarding epsilon and L in the manuscript and rebuttal. In short, I am not fully convinced of the actual benefit of VGF over the vast behaviour-regularized RL methods.
> >
> > [1] Fang, L., Liu, R., Zhang, J., Wang, W., & Jing, B. Y. (2024). Diffusion actor-critic: Formulating constrained policy iteration as diffusion noise regression for offline reinforcement learning. ICLR.

---

> > > ### Author Response · Authors · 2025-11-23
> > >
> > > Thanks for your reply.
> > >
> > > > About inference cost.
> > >
> > > Our main point is that VGF does **not** introduce a significant inference-cost burden compared to existing methods, including both Gaussian-policy and diffusion/flow-policy-based methods, which was the main concern raised by the reviewer. We did not claim that VGF has *better* inference speed than all diffusion/flow-based methods.
> > > Importantly, aside from inference-time cost, VGF benefits from better training speed and lower memory usage due to the removal of the actor, which becomes especially significant in the LLM+RL setting (analogous to GRPO vs PPO). This advantage should not be overlooked.
> > >
> > > >About [1] achieves better performance than VGF on D4RL locomotion tasks.....In short, I am not fully convinced of the actual benefit of VGF over the vast behaviour-regularized RL methods.
> > >
> > > **We want to highlight that that VGF is the only method that consistently perform the best across all behavior-regularized RL domains**. This includes, but is not limited to, offline RL(D4RL mujoco, D4RL antmaze, D4RL adroit, OGBench all datasets) and RLHF tasks. We believe that this is the key contribution of our work and should not be overlooked.
> > > To the best of our knowledge, no existing method achieves such broad generality. Reparameterized policy-gradient methods work well in offline RL but cannot be applied in RLHF settings. Weighted-BC or sampling-based methods work for RLHF, but perform poorly in offline RL tasks.
> > >
> > > In fact, even within offline RL, none single method could work as well as VGF across all different datasets. For example, although [1] performs slightly better than VGF in D4RL mujoco datasets, it falls significantly behind on harder domains such as D4RL Antmaze datasets (VGF: ~80 in large datasets while [1] is ~50) and D4RL adroit datasets (VGF: 100 in pen-human and 106 in pen-cloned while [1] is 81 and 63, respectively). Similarly, FQL performs well on OGBench but poorly on D4RL mujoco and adroit. ReBRAC performs well on D4RL mujoco and adroit but poorly on OGbench datasets.
> > >
> > > > Regarding our claim that "VGF is said to "balance the optimisation conflict between reward maximisation and derivation penalties, avoid brittle coefficient hyperparameter tuning and better preserve the multimodal structure".
> > >
> > > We do acknowledge that this wording may be misleading and will remove the phrase "avoid brittle coefficient hyperparameter tuning" accordingly. However, what remains true is that VGF directly optimizes for the best reward-maximizing policy within a fixed behavior constraint, which aligns naturally with Equation (1). We believe this provides a novel and potentially superior way to balance the optimization conflict between reward maximization and deviation penalties.
> > > In addition, please **do not ignore** other key benefits of VGF, such as: bypassing explicit policy parameterization while still enable multimodal expressivity, enabling adaptive scaling during test-time. Both of which we believe bring significant and meaningful merit to the RL community.
> > >
> > >
> > >
> > > [1] Fang, L., Liu, R., Zhang, J., Wang, W., & Jing, B. Y. (2024). Diffusion actor-critic: Formulating constrained policy iteration as diffusion noise regression for offline reinforcement learning. ICLR.

---

### Official Review · Reviewer_DzjR · 2025-11-01

**Soundness:** 3
**Presentation:** 3
**Contribution:** 3
**Rating:** 4
**Confidence:** 3

**Summary:**

This paper focuses on the area of behavior-regularized reinforcement learning, specifically within the contexts of offline reinforcement learning (RL) and reinforcement learning with human feedback (RLHF). In response to the challenges of training instability and the difficulty of hyperparameter tuning in existing explicit constraint algorithms, the authors propose the Value Gradient Flow. This approach integrates particle-based gradient flow with value functions, utilizing multi-step gradient guidance to iteratively direct the policy towards regions with higher value distributions. Experimental evaluations on offline RL and RLHF datasets, including D4RL and OGBench, demonstrate that this method outperforms existing algorithms.

**Strengths:**

1. The motivation of the paper is clear and straightforward. The writing is fluent and easy to understand, effectively explaining how Value Gradient Flow is integrated into the behavior-regularized reinforcement learning problem.

2. In my view, introducing Particle-based Gradient Flow into the offline reinforcement learning domain and using gradient-based guidance to iteratively shift the behavior cloning action outputs towards higher value regions is an innovative approach. The related experiments also validate the effectiveness of the proposed method.

**Weaknesses:**

1. I suggest that the authors conduct experimental evaluations on more complex environments within the D4RL dataset (e.g., Adroit) as well as on the visual input V-D4RL, to better validate the generalization of the proposed method.

2. In Algorithm 1, the authors present the Q-network learning via TD with the iterated $a^{L\_{train}}\_N$. However, could this lead to overly optimistic learning for the Q-network? In my opinion,  $a^{L_{train}}_N$ is prone to producing out-of-distribution (OOD) actions. What would the results be if the Q-network were first learned using methods like CQL or IQL, and then directly tested?

3. Although the authors emphasize that VGF significantly reduces training costs, the paper lacks quantitative analysis, including but not limited to training time, inference time, and memory usage. Including these details would provide a more comprehensive evaluation of the entire algorithm.

4. I recommend adding a column to each main experiment table to show the results when  $L\_{test} = 0$, which corresponds to the best-of-N sampling method. This would better highlight the improvement brought by VGF, rather than being influenced by ensemble methods like best-of-N sampling or other factors.

5. The citation for Diffusion-QL on line 376 is incorrect.

**Questions:**

Please refer to the "Weaknesses" section, I will raise my score if my concerns are addressed.

---

> ### Author Response · Authors · 2025-11-22
>
> We thank the reviewer for the effort engaged in the review phase and the constructive comments. Regarding the concerns, we provide the detailed responses separately as follows.
>
>
>
> > I suggest that the authors conduct experimental evaluations on more complex environments within the D4RL dataset (e.g., Adroit) as well as on the visual input V-D4RL, to better validate the generalization of the proposed method.
>
> Thank you for the insightful suggestion. We evaluate VGF on more datasets (D4RL Adroit and OGBench visual tasks) as follows, where VGF still shows decent results compared with representative baselines. Note that we originally tested VGF on various tasks under different settings, including D4RL Mujoco, D4RL Antmaze, all OGBench non-visual tasks, and also RLHF tasks, which already include more experiments than what normal published RL papers usually provide.
>
>
> | Algorithm            | BC  | ReBRAC | FQL | VGF |
> |----------------------|-----|--------|-----|-----|
> | pen-human-v1         | 71  | 103    | 53±6  |100±3  |
> | pen-cloned-v1        | 52  | 103    | 74±11  |106±4 |
> | pen-expert-v1        | 110 | 152    | 142±6 |150±1 |
> | door-human-v1        | 2   | 0      | 0±1  |3±2 |
> | door-cloned-v1       | 0   | 0      | 2±1  |3±1 |
> | door-expert-v1       | 105 | 106    | 104±1 |105±1 |
> | hammer-human-v1      | 3   | 0      | 1±1   |2±1 |
> | hammer-cloned-v1     | 1   | 5      | 11±9   |7±4 |
> | hammer-expert-v1     | 127 | 134    | 125±3 |132±4 |
> | relocate-human-v1    | 0   | 0      | 0±0   |0±0 |
> | relocate-cloned-v1   | -0   | 2      | -0±0   |4±3 |
> | relocate-expert-v1   | 108 | 108    | 107±1 |110±1 |
>
>
> | Algorithm                                      | BC  | ReBRAC |FQL | VGF |
> |-----------------------------------------------|-----|-------- |---| ---- |
> | visual-cube-single-play-singletask-task1-v0  | 70  | 83 | 81±12  | 92±6  |
> | visual-cube-double-play-singletask-task1-v0  | 34  | 4  | 21±11  | 60±8  |
> | visual-scene-play-singletask-task1-v0        | 97  | 98 | 98±3  | 98±1  |
> | visual-puzzle-3x3-play-singletask-task1-v0   | 7   | 88 | 94±1  | 98±1  |
> | visual-puzzle-4x4-play-singletask-task1-v0   | 0   | 26 | 33±6  | 52±5  |
>
>
>
>
> > In Algorithm 1, the authors present the Q-network learning via TD with the iterated. However, could this lead to overly optimistic learning for the Q-network? In my opinion, is prone to producing out-of-distribution (OOD) actions. What would the results be if the Q-network were first learned using methods like CQL or IQL, and then directly tested?
>
> For the Q-network, mild conservatism will likely not cause Q-overestimation while allowing more leverage of the generalization of the Q-network, as discussed in previous work such as MCQ (Mildly Conservative Q-Learning for Offline Reinforcement Learning). Also, as we discuss in the paper (lines 266–269):
>
> `One difference in this case is that VGF learns the value function by TD learning (since the train-time flow step is not 0) instead of in-sample learning (Kostrikov et al., 2021b; Xu et al., 2023). We find in practice that TD learning enables better stitching and generalization. This difference makes VGF fundamentally different from Diffusion-based methods (Mao et al., 2024; Frans et al., 2025) that can also do adaptive generation via adjusting the guidance weight but rely on in-sample value learning.`
>
> To support our claim, we provide a comparison in which we replace the value function learned with TD by a value function learned with IQL while keeping all other components unchanged. It can be seen that although VGF with IQL works on some simple Mujoco tasks, using IQL generally fails when it comes to more complex tasks.
>
> | Algorithm    | VGF with TD  | VGF with IQL |
> |---------------|-----|-------- |
> | halfcheetah-m | 57.1  | 52.0 |
> | hopper-m  | 97.9  | 78.8  |
> | halfcheetah-m-r | 49.1  | 43.7 |
> | walker2d-m-r   | 97.8   | 83.6 |
> | antmaze-m-p   | 92.4   | 80.2 |
> | antmaze-l-d   | 83.8   | 54.1 |
> | antsoccer-arena   | 63   | 0.0 |
> | cube-double   | 61   | 0.0 |
> | puzzle-4x4   | 46   | 0.0 |

---

> > ### Author Response · Authors · 2025-11-22
> >
> > > Although the authors emphasize that VGF significantly reduces training costs, the paper lacks quantitative analysis, including but not limited to training time, inference time, and memory usage. Including these details would provide a more comprehensive evaluation of the entire algorithm.
> >
> >
> > Thanks for pointing this out. First, we want to clarify that we did not emphasize that VGF significantly reduces training costs. What we emphasized is that VGF removes the need to parameterize an actor. We now provide the training cost, inference cost, and memory usage of representative baselines across various environments. Specifically, we compare VGF with IQL (Gaussian policy), DQL (diffusion policy), IFQL (flow policy), and FQL (one-step flow policy).
> >
> > The training and inference costs are measured by the time per step under the default set of parameters. Note that VGF uses smaller flow/diffusion steps (5) than Diffusion-QL, so it reduces the inference cost, while we found that Diffusion-QL with fewer steps performs much worse than the original setting.
> > For memory usage, we disable JAX memory preallocation to allow a fair comparison of GPU memory usage.
> >
> > We find that VGF has the lowest training cost and the least memory usage. For inference cost, VGF is consistently faster than DQL and IFQL, while slightly slower than FQL and IQL. We believe it is reasonable to sacrifice a small amount of inference compute to obtain a significant performance boost.
> >
> >
> > | Environment | Metric         | VGF    | Diffusion-QL | FQL    | IFQL   | IQL    |
> > |------------|----------------|--------|---------------------|--------|--------|--------|
> > | halfcheetah-m-e | Training time  | 1.92ms | 2.11ms | 2.10ms | 2.16ms | 2.04ms |
> > | halfcheetah-m-e | Inference time | 0.63ms | 1.18ms | 0.32ms | 0.94ms | 0.36ms |
> > | halfcheetah-m-e | Memory usage   | 484MB  | 496MB  | 498MB  | 498MB  | 498MB  |
> > | antmaze-m-d    | Training time  | 1.94ms | 2.36ms | 2.16ms | 2.22ms | 2.15ms |
> > | antmaze-m-d    | Inference time | 0.71ms | 1.44ms | 0.37ms | 0.93ms | 0.34ms |
> > | antmaze-m-d    | Memory usage   | 488MB  | 500MB  | 500MB  | 500MB  | 498MB  |
> > | humanoidmaze-m    | Training time  | 2.71ms | 3.28ms | 2.78ms | 2.92ms | 3.02ms |
> > | humanoidmaze-m   | Inference time | 0.81ms | 1.54ms | 0.59ms | 0.98ms | 0.47ms |
> > | humanoidmaze-m    | Memory usage   | 640MB  | 648MB  | 654MB  | 654MB  | 652MB  |
> > | hopper-m     | Training time  | 2.18ms | 2.54ms | 2.14ms | 2.21ms | 2.25ms |
> > | hopper-m     | Inference time | 0.67ms | 1.24ms | 0.38ms | 0.88ms | 0.41ms |
> > | hopper-m      | Memory usage   | 484MB  | 494MB  | 496MB  | 496MB  | 492MB  |
> > | walker2d-m     | Training time  | 2.07ms | 2.47ms | 2.20ms | 2.26ms | 2.44ms |
> > | walker2d-m     | Inference time | 0.65ms | 1.24ms | 0.39ms | 0.93ms | 0.39ms |
> > | walker2d-m     | Memory usage   | 484MB  | 496MB  | 500MB  | 498MB  | 498MB  |
> >
> >
> > > I recommend adding a column to each main experiment table to show the results when L_test=0, which corresponds to the best-of-N sampling method. This would better highlight the improvement brought by VGF, rather than being influenced by ensemble methods like best-of-N sampling or other factors.
> >
> > Note that comparing the results of L_test=0 in VGF couldn't disentagle the improvement brought by VGF. This is because in offline RL, even if we set L_test=0, VGF is foundamentally different from best-of-N sampling methods like IDQL since a better Q function is learned due to VGF. Therefore, we would think of varying L_test as a distinct property of VGF since this only comes true when we don't parameterize a policy. And we already provide the investigation and study of this property in Figure 3, we will add more in the final version.
> >
> >
> > > The citation for Diffusion-QL on line 376 is incorrect.
> >
> > Thanks for pointing out that, we will revise it in the revision.
> >
> >
> > Please let us know if any further questions remain. We hope the reviewer can reassess our work with these clarifications.

---

### Comment · Area_Chair_Vxrv · 2025-11-23
**Subject: Follow-up Reviews Required to Proceed**

Dear Reviewers,

The authors have submitted their rebuttal, and we now require your follow-up assessments to move the decision process forward. Please review the authors’ responses and update your evaluations accordingly.

Your prompt follow-up is necessary for us to finalize the meta-review.
Kindly submit your updates as soon as possible.

Best,
Area Chair

---

### Author Response · Authors · 2025-12-03

Dear Area Chair & Reviewers,

Since reviewers are not permitted to participate in the discussion stage, we provide here a concise summary of our paper’s key contributions and clarifications addressing the reviewers’ concerns.

# Contributions:
We introduce **Value Gradient Flow (VGF)**, a fundamentally new paradigm for behavior-regularized reinforcement learning (BR-RL), encompassing—but not limited to—offline RL and RLHF. Compared with parameterized policy-gradient methods, weighted behavior-cloning approaches, and sampling-based methods, **VGF is the only method that consistently achieves state-of-the-art performance across all major BR-RL domains**, including D4RL MuJoCo, D4RL AntMaze, D4RL Adroit, OGBench tasks, and RLHF tasks. To our best knowledge, no prior method demonstrates such broad applicability: reparameterized policy-gradient methods perform well in offline RL but do not extend to RLHF, while weighted-BC and sampling-based methods work for RLHF but work poorly in offline RL.


# Significance:
Our method delivers consistent improvements over prior state-of-the-art baselines, as acknowledged by all reviewers. Beyond performance, VGF offers **several notable advantages**:
1. Elimination of auxiliary behavior regularizers, providing a new and potentially more robust way to resolve the optimization conflict between reward maximization and deviation penalties.
2. Bypassing explicit policy parameterization while retaining strong multimodal expressivity.
3. Enabling adaptive test-time scaling.

We believe these properties make VGF a meaningful and impactful contribution to the RL community.

# Addressing Reviewer Concerns:
1. "VGF increases significant computational cost"

We believe this is not true, We provide comprehensive measurements of training time, inference cost, and memory usage across representative baselines and domains. The results show that VGF benefits from faster training and lower memory consumption due to the removal of the actor—advantages that become especially pronounced in LLM-based RL. For inference, VGF does not introduce substantial overhead relative to Gaussian-policy or diffusion/flow-based policies. We believe that a modest increase in inference compute is a reasonable trade-off for the significant performance gains VGF provides.

2. "VGF appears sensitive to its hyperparameters"

Our ablations show that only the VGF step L and learning rate \epsilon require tuning, and even these are not highly sensitive. Other parameters—including the temperature \alpha—have minimal effect. The pair L and \epsilon jointly control the implicit deviation from behavior particles, analogous to how the explicit regularization coefficient \alpha must be tuned in existing offline RL methods.

3. Concern regarding “removal of the need to balance reward maximization and deviation penalties.”

We agree that our original phrasing was misleading and will remove the claim. Nonetheless, VGF does directly optimize for the reward-maximizing policy without auxiliary regularizers, a formulation that aligns naturally with Equation (1). We believe this represents a novel and promising alternative for handling the inherent trade-off in behavior-regularized RL.

Thank you for your time and consideration.

Sincerely, The Authors

---

### Meta-Review · Area_Chair_zzPj · 2026-01-06

**Summary:**

Reviewers see VGF as an interesting unification of behavior-regularized RL via an optimal-transport / gradient-flow lens, with a practically appealing instantiation (particle transport guided by a learned value function) that spans offline RL and RLHF-style test-time generation. The strongest positive signal is broad empirical coverage (offline + RLHF) and a clear conceptual connection to SVGD/JKO-style updates.
The main reasons the paper trends borderline are not novelty per se, but the credibility of several claims: (i) whether VGF truly reduces tuning burden versus explicit regularizers (it still tunes transport budget and step size), (ii) whether inference/training cost is adequately characterized relative to diffusion/flow baselines, and (iii) whether comparisons against strong recent diffusion-policy methods are framed fairly and transparently.

**Reviewer Concerns:**

Concerns that were addressed in the rebuttal include: adding harder offline domains (Adroit) and visual tasks, providing concrete runtime and memory numbers across baselines, clarifying that VGF uses SFT-initialized particles in RLHF tables, and explicitly softening the misleading “removes the need to balance reward vs deviation” wording. The ethics flag about potential dual submission was also resolved at the AC/PC level with confirmation of no shared authorship, so it should not affect the technical decision.
Concerns still outstanding are mostly about positioning and robustness. Two reviewers remain unconvinced that VGF is meaningfully easier to tune than coefficient-based regularization, since L and epsilon (and sometimes particle count/temperature) still require task-dependent adjustment; this point needs careful, consistent edits throughout the paper. Additionally, the discussion around diffusion-policy baselines (e.g., recent few-step diffusion actor-critic) should be strengthened: either include direct comparisons where feasible or clearly scope claims and explain why performance differs by domain without overgeneralizing.

**Reviewer Scores:**

I expect gCX9 would keep their strong accept score (they explicitly stated concerns were resolved). Reviewer DzjR signaled they would raise the score if concerns were addressed; given the added Adroit/visual results and the cost table, they likely move from borderline reject to around the acceptance threshold.
Reviewer A97B may improve slightly but is unlikely to jump to a clear accept without better framing versus strong diffusion-policy methods and a more honest statement about hyperparameter sensitivity; I would expect them to remain near borderline. Reviewer Y9ry likely stays around marginal accept: the misleading claim was acknowledged and edited, cost was provided, and the ethics concern is cleared, but they still challenge the “suffers less” tuning narrative and may not upgrade unless the camera-ready resolves this cleanly.

---

### Decision · Program_Chairs · 2026-01-26

Accept (Poster)